**Discordance Dating: A New Approach for Dating Alteration Events**

Jesse R Reimink[1*]; Renan Beckman[1]; Erik Schoonover[1]; Max Lloyd[1]; Joshua Garber[1]; Joshua HFL Davies[2]; Alexander Cerminaro[1,3]; Morgann G Perrot[2,4]; Andrew Smye[1]

[1]Department of Geosciences, The Pennsylvania State University, University Park, PA, USA
[2]Département des sciences de la Terre et de l'atmosphère / GEOTOP, Université du Québec à Montréal, Montréal, CA
[3]Department of Geosciences, Texas Tech University, 1200 Memorial Circle, Lubbock, TX, USA
[4]Department of Earth and Planetary Sciences, McGill University, Montréal, CA
*corresponding author: jreimink@psu.edu

**Abstract:**
Zircon is the premier geochronometer used to date igneous and metamorphic processes, constrain sediment provenance, and monitor key events in Earth history such as the growth of continents and the evolution of the biosphere. Zircon U-Pb systematics can be perturbed by the loss or gain of uranium and/or lead, which can result in disagreement between the apparent radiometric ages of the two U-Pb decay systems – a phenomenon that is commonly termed 'discordance'. Discordance in zircon can be difficult to reliably interpret and therefore discordant data are traditionally culled from U-Pb isotopic datasets, particularly detrital zircon datasets. Here we provide a data reduction scheme that extracts reliable age information from discordant zircon U-Pb data found in detrital zircon suites, tracing such processes as fluid flow or contact metamorphism. We provide the template for data reduction and interpretation, a suite of sensitivity tests using synthetic data, and ground-truth this method by analyzing zircons from the well-studied Alta Stock metamorphic aureole. Our results show accurate quantification of a ~24 Ma in situ zircon alteration event that affected 1.0-2.0 Ga detrital zircons in the Tintic quartzite. The 'discordance dating' method outlined here may be widely applicable to a variety of detrital zircon suites where pervasive fluid alteration or metamorphic recrystallization has occurred, even in the absence of concordant U-Pb data.

**1. Introduction:**
Discordant zircon U-Pb data may be created in multiple ways, including pure Pb-loss (Nasdala et al., 1998), uranium addition (Garber et al., 2020; Grauert et al., 1974; Seydoux-Guillaume et al., 2015), a combination of both U gain and Pb loss (Andersen and Elburg, 2022), clustering of Pb into nanoparticles (Kusiak et al., 2015, 2023), mixing between two domains of different ages (Schoene, 2014 and references therein), and partial metamorphic recrystallization (e.g., Davis et al., 1968; Geisler et al., 2007; Hoskin and Black, 2002). Experimental evidence has shown that during the radioactive decay of U and Th, alpha recoil damage accumulates in the crystal lattice and can lead to interconnected pathways allowing for chemical disturbance leading to discordant U-Pb dates (Geisler et al., 2002, 2007; Salje et al., 1999; Trachenko et al., 2002). Thus, many geological processes are known to induce discordance, including the buildup of metamict domains inside zircon crystals (Nasdala et al., 1998), meteorite-impact induced shock effects (Moser et al., 2009, 2011), crystal-plastic deformation (Reddy et al., 2006), fluid induced dissolution-reprecipitation (Geisler et al., 2002, 2007), weathering processes (Andersen and Elburg, 2022; Pidgeon et al., 2016), pyrometamorphic heating (Ulusoy et al., 2019), and metamorphic

recrystallization and overgrowth that inherits radiogenic Pb during recrystallization (Mezger and
Krogstad, 1997; Schoene, 2014).
The analytical expression of discordance has long been utilized for understanding the events that
cause it, yielding lower intercept ages interpreted to be resetting/alteration events. This is typically
done using linear regressions (Davis, 1982; Vermeesch, 2021; York, 1968) through datasets that
fall along a single chord – meaning they have one single upper intercept event (often igneous or
metamorphic crystallization) and one single lower intercept event (caused by one of several
potential mechanisms, discussed above). Such an approach has yielded meaningful ages for a
variety of geological processes (Mezger and Krogstad, 1997; Moser et al., 2009, 2011; Schoene,
56 2014).
While most modern U-Pb studies aim to minimize discordance rather than use it (see Schoene,
2014, and references therein), there has been increasing focus on using discordant data to date
geological events, as discussed below.  Sedimentary detrital zircon analyses present an opportunity
for dealing with discordance when compared to igneous and metamorphic zircon studies (e.g.,
Kirkland et al., 2017; Morris et al., 2015; Reimink et al., 2016). Because each detrital zircon in a
sediment shares the general post-depositional history with other detrital grains, discordant data
may be interpreted together. Without the constraint of a geological history, discordant U-Pb data
can be ambiguous or impossible to interpret. When dealing with sedimentary detrital zircon U-Pb
data, other assumptions may be drawn upon to related discordant zircon U-Pb data points to one
another, leveraging variable discordance to extract geologically meaningful age information from
the full population. One assumption that is shared by most studies investigating discordant detrital
zircon data is the assumption that zircons within a sedimentary rock might experience the same
geological events (fluid flow, metamorphism, etc.) that could variably affect the U-Pb systematics
of individual zircon grains to generate a spread of discordant data. This shared history, but with
variable imprints, can be very useful when attempting to infer geological events that may impose
discordance.
Several suites of studies have developed methods investigating discordant zircon U-Pb data. One
such suite relies on a comparison between discordant and concordant portions of detrital zircon
populations (Kirkland et al., 2017, 2020; Morris et al., 2015; Olierook et al., 2021), the so-called
"Concordant-Discordant Comparison" (Kirkland et al., 2017). This method requires a concordant
data population to serve as a comparator and then goes on to calculate an expected upper intercept
age distribution from discordant data points using a range of lower intercept ages. The calculated
upper intercept age distribution is then compared to the concordant data distribution (assumed to
be "true" distribution) using a K-S test or other 'similarity' metric, from which the most likely time
of discordance is then inferred. This method requires that concordant and discordant data subsets
be derived from very similar underlying populations, and that discordance-inducing processes
affect all populations of grains equally such that no biases are imparted on the discordant
population distributions. Statistical tests can then be used to evaluate the merits of this assumption.
Another approach (which is built upon in the present work), maps probability density from a given
dataset that falls along predefined chords in U-Pb space (Reimink et al., 2016). This results in a
'likelihood' map across a range of upper and lower intercept ages. This likelihood map can be
interrogated many ways, including evaluating upper intercept probability across a range of lower

intercepts, or the inverse. Additionally, discordant and concordant data points can be weighted to focus on discordant probability and suppress the effect of clusters of concordant data on the likelihood map. This method inherently incorporates analytical uncertainty by uniquely calculating the probability contributed to each data point to a given chord (Davis, 1982). Thus, uncertain data points contribute small probability densities to many chords, whereas highly precise data contribute large probability densities to fewer chords.

Additionally, discordant data generated by multi-phase mixing of growth zones have also been utilized to reconstruct upper intercept ages of core components in several ways. These methods simply recalculate upper intercept ages using a single lower intercept age, projected through discordant data. This is an output of the likelihood mapping discussed above (Reimink et al., 2016), and such a calculation has used when the a lower intercept likelihood peak is determined to be an igneous overgrowth event due to anataxis (Rasmussen et al., 2023). Additionally, an inverted CDC approach has been used when the rim growth ages are independently determined (Olierook et al., 2021), by reconstructing the upper intercept ages by using the mean of a data point and an the lower intercept age.

Other approaches to treat discordance provide mechanisms for assessing or filtering for discordant populations (Andersen and Elburg, 2022; Powerman et al., 2021; Vermeesch, 2021), and identifying Pb-loss specifically in igneous samples (Sharman and Malkowski, 2024). At times filtering of discordant data can result in a significant number of data being discarded, at times up to more than 60% of the total dataset (e.g., Clemens-Knott and Gevedon, 2023).

Despite the difficulty in linking the ages between individual discordant zircon in detrital sediments, one useful assumption can safely be applied: after the deposition of the sediment, all the zircon grains have a similar thermal and geological history. However, importantly, each grain will respond to these geologic events differently due to the unique crystallization, radiation damage, and previous thermal/annealing history of that particular grain. This shared history, but variable response, can be utilized to estimate ages of lower-temperature events than are typically recorded in zircon U-Pb ages (Kirkland et al., 2017; Morris et al., 2015; Reimink et al., 2016). We provide a robust analytical framework for determining the timing of these discordance-inducing events. We empirically show that detrital zircon grains from a sediment that was affected by contact metamorphism and fluid flow can be used to estimate the age of metamorphism and fluid alteration using zircon U-Pb discordance alone, without a significant concordant data fraction (c.f., Kirkland et al., 2017; Morris et al., 2015). Discordance is an undervalued and useful feature of detrital zircon populations, though more work is required to fully evaluate the applicability of the various frameworks that use discordance to date geological events.

## 2. Methods
### 2.1 Theoretical Framework
Here we date in situ alteration of detrital zircons based on the approach to discordance outlined by Reimink et al. (2016). In this framework, a numerical algorithm is used to calculate the probability distribution of a zircon U-Pb dataset across a range of synthetic discordia chords that represent candidate chords spanning a defined age range. Essentially, a 'mesh' of potential discordia chords is created at a defined interval (e.g., every 1 Ma a new chord is created) wherein each line has a unique upper and lower intercept age (Fig. 1b). Then, the total probability that falls on each chord

is calculated by determining the probability that each individual datapoint contributes to each
chord, using the equations of (Davis, 1982). The total probability is summed for each chord and
termed 'likelihood'. Further details on the method can be found in Reimink et al. (2016).

Reimink et al. (2016) discussed several different
normalization strategies to avoid biasing the
calculated probability distributions, including
homogenizing the uncertainty across a U-Pb dataset,
weighting against concordance (towards discordance),
and others. These normalizations can be useful to
prevent artificially biasing due to clusters of
concordant data, or biasing due to heteroscedastic data
(i.e., with non-uniform variance; Vermeesch, 2012). In
detrital zircon datasets, collected from rocks without
post-depositional disturbances, the chord with the
highest probability is likely to be associated with a
group of zircon crystals that have the same upper
intercept age, regardless of whether they are
discordant or not. In such an analysis, the upper
intercept ages are the most useful output. Thus, dating
a post-depositional discordance-inducing *lower-
intercept* event requires a different approach. The
present method modifies the original calculations to
determine the most likely lower intercept age across a
sample set that may have a range of upper intercept
ages.  For example, using the Reimink et al. (2016)
calculation on a detrital sample which has a large
population of near-concordant grains at 1600 Ma and
experienced a Pb loss event at 30 Ma, the chord with
the highest probability would likely be chord between
30 and 1600 Ma (Fig. 1). However, 30 Ma lower
intercept ages associated with any other upper intercept
age would not be included in the probability of the 30
Ma lower intercept age, as only the 30−1600 Ma chord
is considered. If a detrital zircon population contains
grains that crystallized at 1000 Ma, 1200 Ma and 2700
Ma and these grains also experienced *in situ* Pb-loss at
30 Ma, that Pb-loss would go mostly undetected by the
previous method, though each upper intercept age
would be resolved independently.

To rectify this issue, here we introduce a modification
to the original calculation. We add an additional step

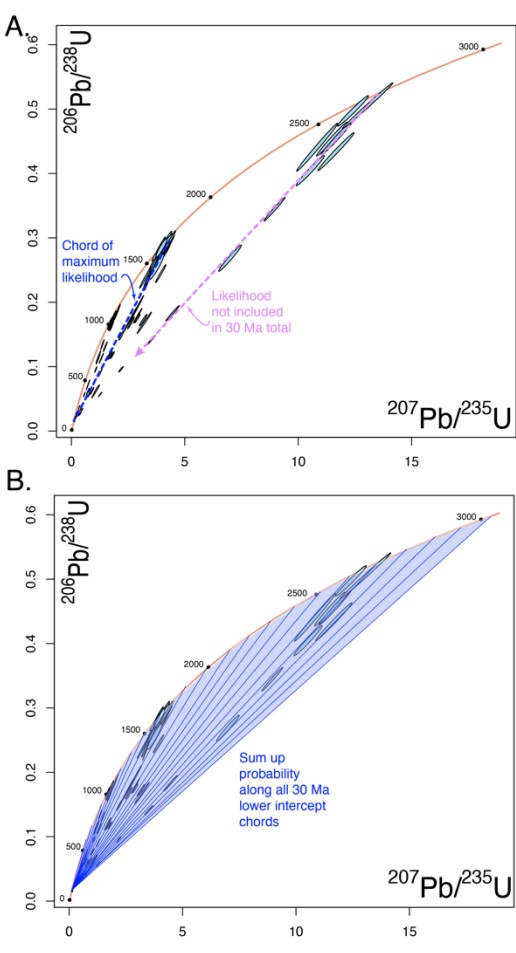

Figure 1: Example data showing why the sum probability density is required to accurately define likelihoods for lower intercept ages. In this synthetic dataset, there are many near-concordant ~1600 Ma datapoints. This cluster of near-concordant data will yield a higher probability for any chord anchored at 1600 Ma. A new method is required to evaluate the total probability contributed to any lower intercept age from a range of upper intercept ages. Panel B shows how the total probability density contributed to all lines with a 30 Ma lower intercept age could capture the likelihood of lower intercept ages in that window.

that sums the total probability aggregated to each lower intercept age. Using the example above,
we add up all the probability accrued to all the lines with a lower intercept age of exactly 30 Ma.
Thus, the probability accumulated by the 30−1580 Ma chord would be added to the probability
accumulated by the 30–1581 Ma chord, the 30–1582 Ma chord, etc. This sum is then divided by
the number of chords that have a given lower intercept age to normalize across the age range of
interest. This value is then termed 'summed likelihood' as it is a normalized value and no longer a
probability density estimate. The results of this modeling approach theoretically return an estimate
of the potential that a given lower intercept age may be a true post-depositional disturbance age,
though it is normalized by the number of analyses and number of chords in any given model.
**2.2 Benchmarking with Synthetic Datasets**
To test the sensitivity and accuracy of our theoretical approach to extracting the ages of post-
depositional discordance-inducing events from detrital zircons, we constructed several synthetic
datasets and benchmarked our approach against single isochron regression techniques. We created
synthetic data that aims to replicate aspects an actual zircon U-Pb dataset from Tintic formation
detrital zircons (presented in Section 3). To do this, we used three categories of synthetic datasets
(Fig. 2).
1. A U-Pb dataset that defines a perfect discordia line with an upper intercept of 1800 Ma a
lower intercept of 30 Ma
2. A dataset with three upper intercept ages (1800 Ma, 1400 Ma, and 1100 Ma) all of which
have discordance imposed at a shared lower intercept of 30 Ma
3. A dataset where each grain has an upper intercept age randomly selected from a range of
ages (1800–1100 Ma) and all have a shared lower intercept of 30 Ma.

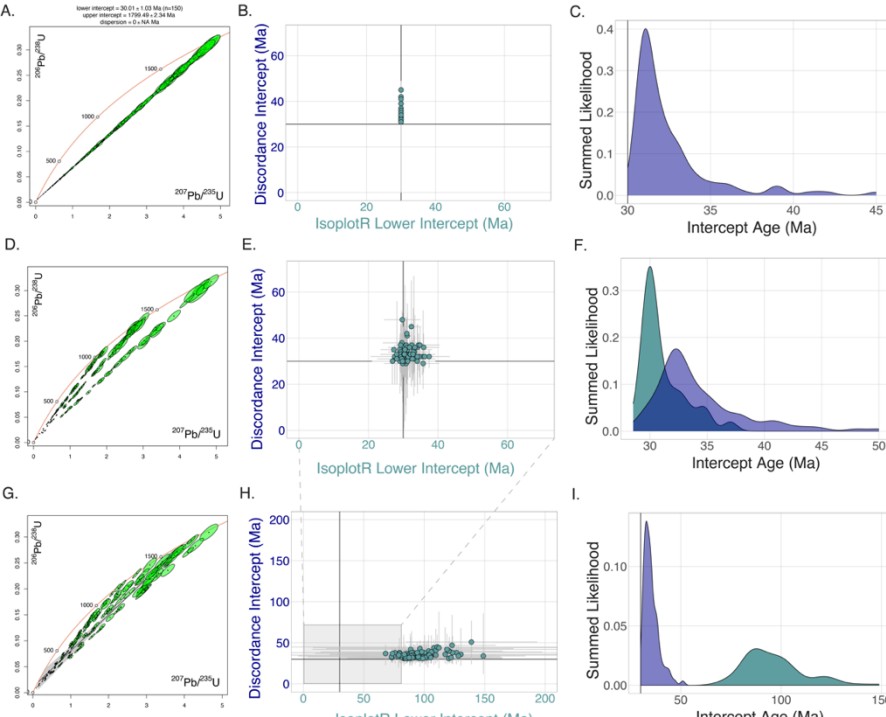

For each of the three
categories, a synthetic data
population was generated
that contained 150 U-Pb data
points. The number of
datapoints was chosen as a
typical number of analyses
in detrital zircon datasets.
However, see Section 2.3 for
further discussion of this
aspect. Each data point was
randomly assigned an upper
intercept and lower intercept
age based on the model
structure. In the case of the
perfect line, all data points
had the same upper intercept
age. Then, each data point
was assigned a random
amount of discordance
between 99.95% and 1%,
selected from a flat

*Figure 2: Input data and results from a synthetic data modeling procedure. Panels A, D, and G show examples of the synthetic input data, plotted on U-Pb concordia diagrams. The center panels (B, E, H) show the mean and uncertainties of the Isoplot R regression lower intercept age compared to the lower intercept values derived from the discordance modeling approach. Each data point represents a single synthetically generated dataset of 150 data points. The rightmost pane (C, F, I)l shows the distribution of lower intercept ages from each method with the blue curve representing the ages produced by the discordance model and the green curve representing the ages generated by the model 1 regression in IsoplotR. Note the change of scale in Panel H compared to E and B.*

probability distribution. This random discordance accounts for the fact that each individual zircon
grain, and indeed portions of grains, have differing resetting susceptibilities due to a variety of
factors. The ratios of interest for each data point were assigned a random uncertainty value between
2–5% of their isotopic composition.

Each synthetic data set of 150 points was then evaluated in two ways. First, the traditional U-Pb
regression ages were derived using the commonly used "York fit" approach (York, 1968) as
implemented in IsoplotR (Vermeesch, 2018b), from which the upper and lower intercept ages were
extracted. For the model constructed with three discrete upper intercept ages, we calculated three
independent isochron regressions
in IsoplotR, one for each
population of synthetic data with a
unique upper intercept age, and
then calculated the weighted mean
lower intercept age of these three
regressions. Second, the data were
input into our discordance
modeling procedure and the
maximum sum likelihood was
extracted as well as the full width,
half max of that peak – a measure
of the spread in our modelled
spectral data. This entire procedure
was repeated 100 times for each of
the three categories, resulting in
300 model runs.

The results from the outputs of the
'York fit' approach (model 1
regression in Isoplot R) are
compared to our discordance
modeling approach to benchmark
the discordance dating procedure
against well-accepted regression
methods. Figure 2 shows the three
different types of models, with the

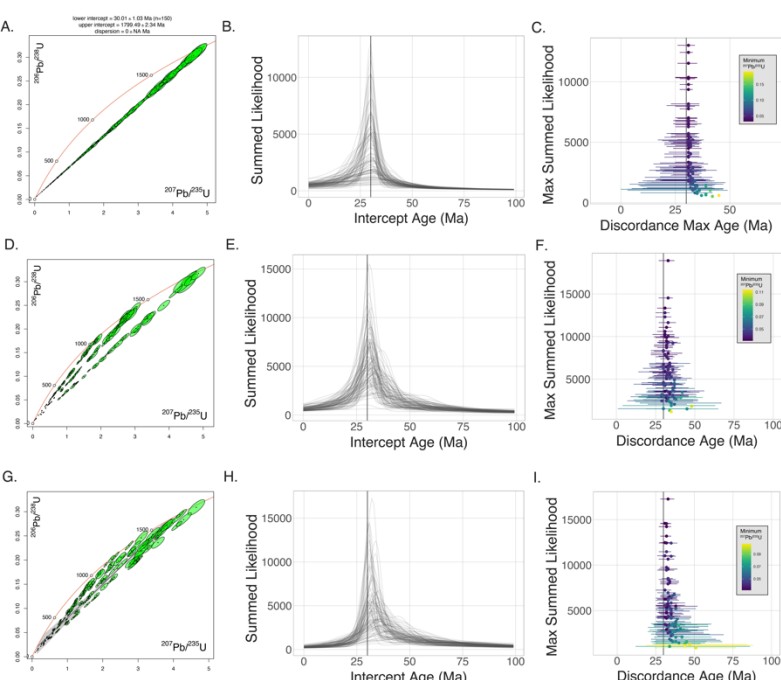

Figure 3: Results from the three sets of model runs for the discordance modeling procedure. The left panels show examples of the input datasets (same as Fig. 2). The central panels show lower intercept likelihood curves for the 100 model runs in each category of model. The right panels show the peak location, the uncertainties (derived by a full-width half-max calculation), and are colored according to the lower $^{207}Pb/^{235}U$ value in a given synthetic dataset. The discordance modeling procedure yields similar results independent of the input data type.

calculated lower and upper intercept ages from each synthetic dataset. Note that in Fig. 2C, the
IsoplotR lower intercepts are tightly bound at 30 Ma so the probability density is not shown here.
Figure 3 shows the discordance dating outputs in more detail, with individual age distributions
shown from the discordance dating outputs.

The outputs of each calculation method are summarized in Fig. 4. For the simplest scenario, the
perfect discordia line, the discordance dating model has larger uncertainties associated with the
lower intercept age as compared to York-fit regressions. Note that in this version of the analysis,
we are focusing on a small portion of the total U-Pb Discordia space, such that the area under each
individual curve in Fig. 3C, F, and I are not all uniform. Sharp peaks are correlated to how close
the 'youngest' analysis is to the 30 Ma lower intercept, where precise lower intercept estimates are
typically derived from populations of data points that have some analyses close to the concordia
curve near the lower intercept age. It is expected – and observed (Fig. 4) – that the York fitting
procedure should outperform the discordance dating algorithm with respect to precision on the
lower intercept for the simplest case where
data can be regressed on a single line.
However, both methods still recover the
accurate lower intercept age within
uncertainty.
When considering more complex U-Pb
populations (for example the lower panels
in Fig. 4) that contain many primary (upper
intercept) U-Pb ages, the discordance
dating procedure expectedly outperforms
single linear regression and weighted
means of multiple linear regressions, as
used in Isoplot calculations. In the case of
having multiple discrete upper intercept
ages (middle panel in Fig. 4), the
discordance dating, although less precise,
provides improved accuracy and likely a
more reliable uncertainty estimate.

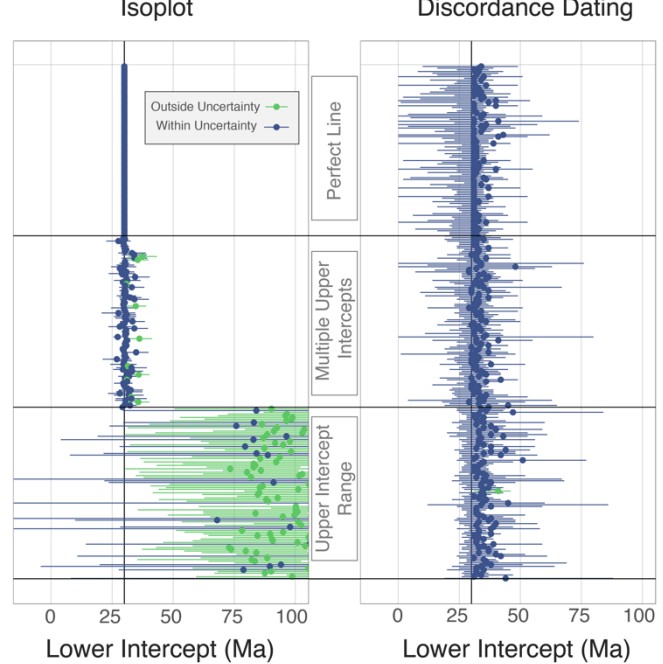

Figure 4: A comparison of the lower intercept ages and uncertainties for the Isoplot regression calculations and the Discordance Dating method for all synthetic data runs. Each point is a unique dataset and the uncertainties are given for each regression point. The color of each point corresponds to whether the lower intercept age is within uncertainty of the pre-defined lower intercept of 30 Ma or not. The Discordance dating method provides more reliable results across a range of data types than simple linear regression models. d

This is an expected result. A linear
regression method is obviously poorly
suited for highly scattered data that do not
share a common linear relationship.
Another method for extracting lower-
intercept age information, the
Concordant-Discordant Comparison
method (Kirkland et al., 2017; Morris et al., 2015) requires a fraction of concordant data to use as
a comparator for refining a lower-age estimate. Due to this requirement, the CDC technique is not
expected to perform well in these synthetic datasets that have very little concordant data, and
indeed it does not perform well on the natural Tintic detrital zircon data discussed later (see Section
311    3.3).

This exercise also sets expectations for the uncertainties and potential biases on the lower intercept
that can be derived from this procedure. Focusing on the models with a range of upper intercept
ages, discordance dating returns a maximum uncertainty envelope that is on average ~20 Ma
(average peak width using FWHM on each peak). However, the central peak of the discordance
dating (the maximum likelihood lower intercept age) for these 100 models is 35 Ma +/- 6.8 Ma
(2SD), which may be a better uncertainty estimate for the discordance dating procedure (discussed
in Section 3.2 for natural data). The discordance dating procedure rarely returns peak centers that
are younger than 30 Ma, though it commonly returns peak centers older than 30 Ma. This bias to
older lower intercept ages is likely due to the sensitivity of the discordance dating procedure to the

position of the youngest point in any dataset – or the most 'reset' grain in any population. This is also important for our later interpretations – discordance dating is highly unlikely to return a maximum peak position that is too young (assuming no modern-aged discordance in the population), and any maximum peak position can likely be interpreted as a maximum age value for the true lower intercept resetting age.

**2.3 Age Sensitivity and Precision**

The underlying modeling approach used in this work fundamentally relies on an angular intersection between discordant data arrays and the 'concordia' curve in U-Pb space. Thus, due to the shorter half-life of the $^{235}$U isotope system, this angularity will decrease as a function of the maximum age of the oldest zircons in any sample set. Put another way, older detrital zircon grains might be more likely to yield precise discordance dating lower intercept ages.

In an attempt to quantify the effect of older primary ages on discordance dating intercept age precision and provide guidance on the geological scenarios where discordance dating might provide useful chronological information, we conducted an additional suite of simulations that varied the primary age range of zircons in synthetic detrital zircon datasets. These simulations followed the methods for the datasets shown in Fig. 3G-I. We randomly selected a maximum upper intercept age between 100 Ma and 3500 Ma, then randomly selected a minimum upper intercept age that is older than 100 Ma, but younger than the maximum upper intercept age. We then followed the same steps outlined above, randomly generating datapoints with upper intercepts in that age range, randomly applying discordance, and then running the discordance dating methodd. We repeated this process 100 times with varying maximum and minimum upper intercept ages, all containing induced discordance at 30 Ma.

Results from this modeling exercise are
somewhat non-intuitive (Fig. 5). Within
the confines of the model parameters (an
equal number of datapoints,
homogeneous discordance distribution,
random sampling, etc.) there is limited
correlation between the precision in any
discordance dating lower peak and the
oldest age in a detrital zircon population
(Fig. 5D). Much of the variation in
discordance dating lower intercept ages
seems to still be explained simply by the
amount of discordance in the data – the
more discordant the data the more
precise any discordance dating result is
likely to be (Fig. 5E). There does not
appear to be a younger limit on the
applicability of the discordance dating
procedure, nor any obvious constraints
on the upper age range of a detrital zircon
dataset. In our synthetic models there is
a similar range of peak distributions, a
similar structure of the underlying
likelihood distributions, and little excess
imprecision induced by having younger
grains in the sample. Thus, a
geochronologist might hope to gain
some degree of lower intercept age
insight from the discordance dating
method across a wide range of sample
types, including samples with
exclusively Phanerozoic detrital zircons.
However, this modeling reinforces the
need for discordant data to produce

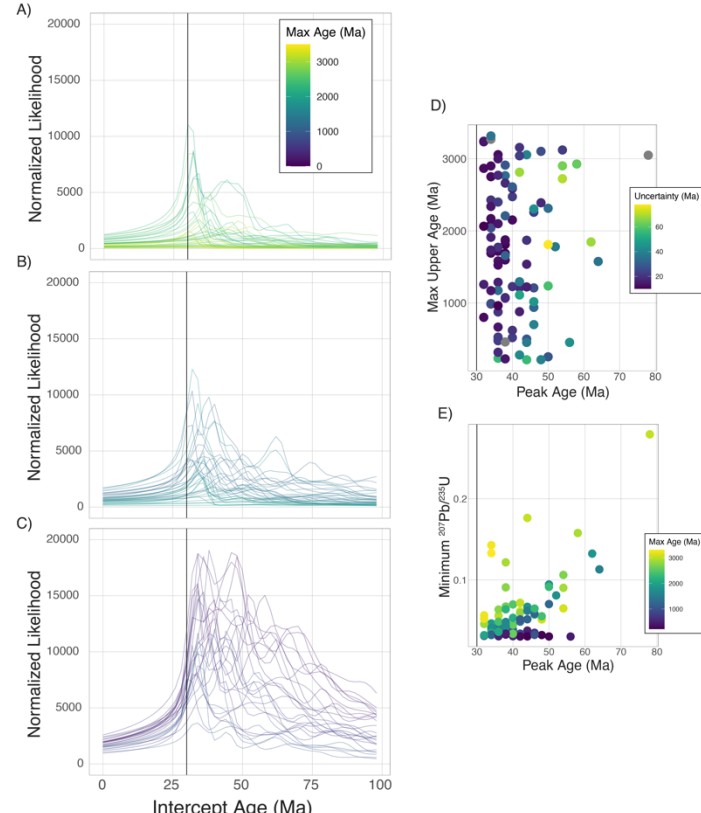

Figure 5: Results from synthetic models that varied the maximum primary age of the detrital zircon population, with maximum ages ranging from 3500 Ma to 200 Ma. Panels A, B, and C show the discordance dating outputs for datasets with maximum ages greater than 2 Ga, maximum ages between 1 and 2 Ga, and maximum ages below 1 Ga, respectively. Panel D shows the summary of the maximum lower intercept peak age plotted against the oldest upper intercept grains in the population, with little correlation between them. Panel E shows a general correlation between the minimum $^{207}Pb/^{235}U$ in the discordant dataset and the resulting discordance dating peak age, reaffirming the conclusion that the more discordant individual analyses are in a given dataset, the more accurate discordance dating can be.

reliable results. Thus, an optimal analytical campaign will intentionally analyze discordant
portions of zircon grains that would be typically avoided in modern in situ analytical routines.
Importantly, enough analyses are required, likely in the many dozens of discordant datapoints, to
achieve a sufficient data density along meaningful discordia arrays.
**3. Alta Stock Detrital Zircon Results**
*3.1 Methods and Results*
To test the discordance dating strategy outlined above on natural samples with a known history,
we conducted a laser ablation ICPMS analytical campaign targeting a well-studied metamorphic
aureole with a known fluid-alteration history. We sampled a metasedimentary unit from the well-
studied Alta stock metamorphic aureole, Utah (Brenner et al., 2021; Cook et al., 1997; Moore and

Kerrick, 1976; Stearns et al., 2020). The Alta stock intruded a suite of mid-Paleozoic sedimentary rocks in the late Paleogene (36–30 Ma; Stearns et al., 2020 and references therein) and was exposed through Miocene-aged uplift and titling along the Wasatch fault and Pleistocene glaciations and erosion in Little Cottonwood Canyon (Stearns et al., 2020 and references therein). The Alta stock is one of a handful of plutons in the region that formed between ~36 Ma and ~30 Ma (Bromfield et al., 1977; Crittenden et al., 1973; Kowallis et al., 1990; Stearns et al., 2020). Titanite and zircon age constraints from the Alta stock indicate it was emplaced near the early stages of this interval (Crittenden et al., 1973; Stearns et al., 2020). Stock emplacement was accompanied by contact metamorphism in the host units and hydrothermal fluid alteration (Brenner et al., 2021; Cook et al., 1997; Moore and Kerrick, 1976). This fluid flow was primarily down-temperature and was focused on mixed carbonate-siliciclastic beds facilitated by porosity-forming decarbonation reactions (Cook et al., 1997). Previous geochronology on magmatic and overprinted zircon along with magmatic and metamorphic aureole titanite phases indicates Alta stock contact metamorphism extended from >30 to 25 Ma (Stearns et al., 2020). This, in combination with trace-element thermobarometry, led Stearns et al. (2020) to conclude that the earliest (36–30 Ma) phases were dominated by high temperature plutonism and metamorphism, whereas hydrothermal alteration activity remained active until ~23 Ma, particularly at the margins of the Alta stock.

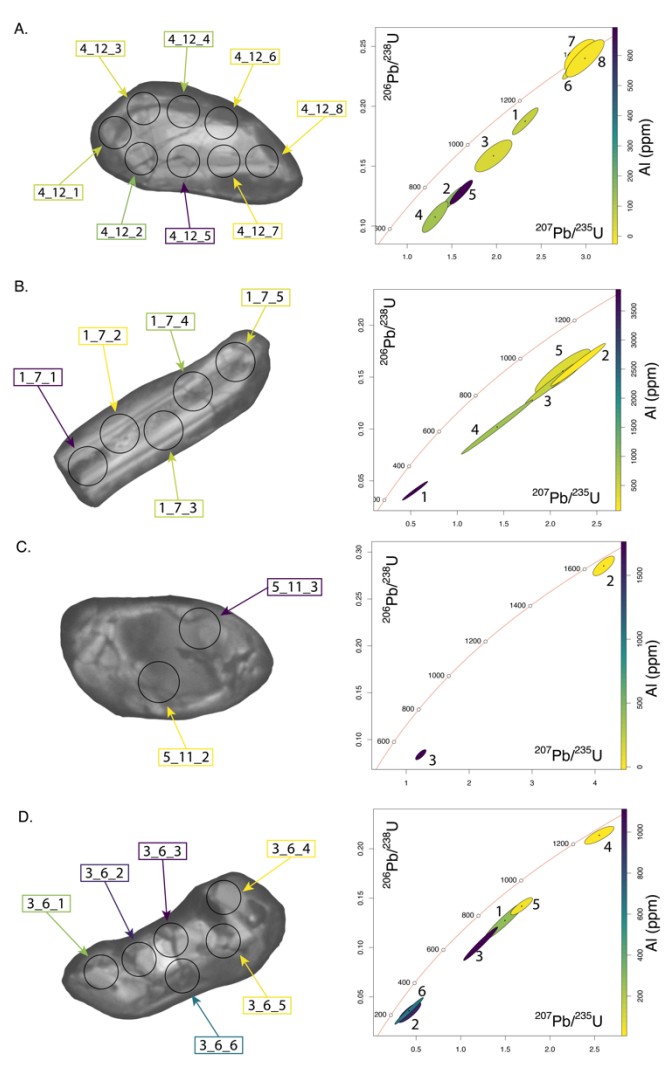

Figure 6: Representative Tintic detrital zircons and their U-Pb isotopic data. Each analysis point is labeled, where color labels correspond to the Al concentration of the analytical volume. The color scales vary between grains. Spot sizes are 30 µm in diameter.

We focused on a single sample of the Cambrian Tintic quartzite. This was collected ~200m from the contact with the Alta stock contact, above the tremolite-in isograd, and experienced metamorphic and fluid-alteration temperatures of ~450° C (Brenner et al., 2021; Cook et al., 1997). This metamorphic and alteration history, combined with the ages of detrital zircons found in the Tintic formation ranging from 1.0 Ga to <2.5 Ga (Matthews et al., 2017), provides an ideal testing ground for determining the accuracy and precision of the discordance dating technique.

Detrital zircons were isolated using standard crushing and mineral separation practices. Limited hand picking of zircons was conducted in an attempt to avoid biasing the results at that stage. Zircons were selected from other heavy minerals, but no preference was given for quality or morphology of zircon grains at this picking stage. Zircons were mounted in epoxy, polished to midsection, imaged using secondary electron imaging techniques, and subsequently analyzed by laser ablation techniques following the methods of (Cipar et al., 2020; Schoonover et al., 2024). A Teledyne/Photon Machines Analyte G2 excimer laser was used, with a Helex2 ablation cell, for all laser ablation work.

Zircon U-Pb and trace element data was collected in three sessions from Feb. 2023 to May 2023. Two sessions, on Feb. 28th, 2023 and March 3rd, 2023, collected U-Pb-TE on the Thermo Scientific iCapRQ quadrupole mass spectrometer in the LionChron analytical facility. Isotopes measured included $^{27}$Al, $^{29}$Si, $^{44}$Ca, $^{51}$V, $^{57}$Fe, $^{146}$Nd, $^{147}$Sm, $^{163}$Dy, $^{172}$Yb, $^{204}$Pb, $^{206}$Pb, $^{207}$Pb, $^{232}$Th, and $^{238}$U. The analytical session on May 30th, 2023 utilized split stream techniques, where $^{204}$Pb, $^{206}$Pb, $^{207}$Pb, $^{232}$Th, $^{235}$U and $^{238}$U were analyzed on the Thermo Scientific Element XR ICPMS system and $^{7}$Li, $^{23}$Na, $^{27}$Al, $^{29}$Si, $^{31}$P, $^{43}$Ca, $^{45}$Sc, $^{49}$Ti, $^{55}$Mn, $^{57}$Fe, $^{59}$Co, $^{60}$Ni, $^{85}$Rb, $^{88}$Sr, $^{89}$Y, $^{90}$Zr, $^{93}$Nb, $^{119}$Sn, $^{133}$Cs, $^{137}$Ba, $^{139}$La, $^{140}$Ce, $^{141}$Pr, $^{146}$Nd, $^{147}$Sm, $^{153}$Eu, $^{157}$Gd, $^{159}$Tb, $^{163}$Dy, $^{165}$Ho, $^{166}$Er, $^{169}$Tm, $^{172}$Yb, $^{175}$Lu, $^{180}$Hf, and $^{182}$W were measured on the iCapRQ quadrupole mass spectrometer. $^{235}$U was calculated from $^{238}$U and the U-isotopic composition of 137.818 (Hiess et al., 2012) due to low $^{235}$U signals. NIST SRM 612 glass was used as a trace-element primary reference material and zircon 91500 was used as a primary reference material for U-Pb isotopic measurements. Uranium-lead and trace element data was filtered, standardized, and normalized using the Iolite data reduction software.

Resulting U-Pb isotopic data for zircon reference materials shows good agreement with accepted values across all three analytical sessions (see "ZirconRM_UPbCompilation.png" in the online repository). The resulting $^{206}$Pb/$^{238}$U ages generally within uncertainty of the accepted values, apart from: GJ-1 from Sessions #2 and #3 (~3% too low), and Peixe from Session #1 (~3 % too high). However the results are accurate enough for the purposes of this study, which relies on U/Pb variation in across a much larger range of U-Pb space.

Trace element data from the zircon reference materials similarly shows consistency within the documented range of trace element concentrations in zircon reference materials (see "ZirconTE_Supplementary.pdf" in the online repository). For instance, the Al concentrations in GJ-1 are nearly identical to published values, with a mean of 3.8 compared to an accepted value of 3.75 ppm (Caulfield et al., 2025). Importantly, though the Al concentration in some reference materials are well characterized (Caulfield et al., 2025), other elements typically used as indicators of alteration (Ca, Fe) are not. Even within a well-studied reference material such as 91500, Al concentrations vary substantially intra- and inter-grain. Though some of our reference material trace-element concentrations are accurate, the concentrations of Al, Ca, and Fe may not be. However, we have chosen to include the concentrations of these alteration elements in our data tables to give the reader a rough estimate of the expected concentrations in altered zircon domains. Further work on the chemical implications of alteration is needed to help characterize the various potential causes of discordance, and we hope that broad estimates of the concentrations of such elements in altered zircon grains can help guide such future work.

Secondary zircon reference materials include Peixe, GJ-1, Plesovice, and MudTank. Secondary reference material U-Pb results are shown in the supplements. The only session where a secondary reference material's U-Pb age was > 2% outside of the accepted age was in the March 3rd session, where GJ-1 had an average $^{206}Pb/^{238}U$ age ~3% lower than the accepted age. However, during the same session Peixe returned a $^{206}Pb/^{238}U$ age in line with the accepted age. Position-dependent fractionation may have played a role in the slightly increased inaccuracy of the GJ-1 zircon during this analytical session. All other sessions returned secondary reference material results within 2% of the accepted ages (see supplementary materials for data and results).

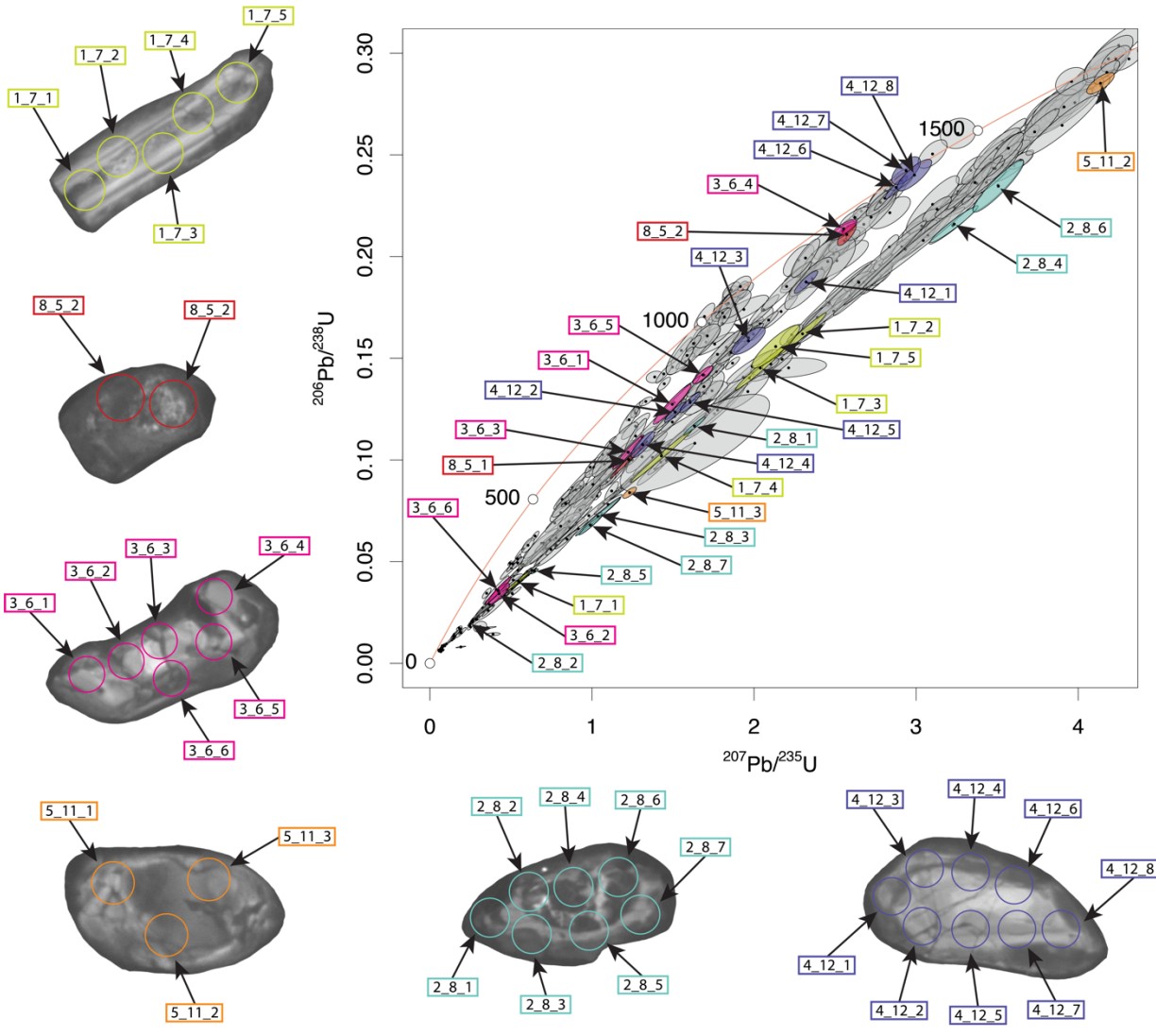

*Figure 7: U-Pb data from detrital zircons extracted from the Tintic quartzite sample studied in this work. Zircon secondary electron microscope images are shown with spot locations correlated to individual data points. For scale, laser spots are all 30 μm.*

Detrital zircons from the Tintic formation show a wide range of igneous and metamorphic/fluid alteration textures. Many (Fig. 6a,b) contain internal zones with apparent oscillatory zoning while others (Fig. 6a,c,d) show clear metamorphic rims along with wispy internal disturbance textures

(Corfu, 2003). While the full dataset contains a wide range of U-Pb isotopic discordance (Fig. 7),
individual grains can often display a wide range of internal age and chemical variability (Figs.
497    6,7).


Zircons from this sample show a range of $^{207}$Pb/$^{206}$Pb ages between 1 Ga and 1.8 Ga, and a wide
range in $^{206}$Pb/$^{238}$U ages (Figs. 6,7), leading to a large spread in the associated discordance. In
typical detrital zircon work, where filtering for concordance is normal (e.g., Gehrels, 2014), the
vast majority of these analyses would be filtered out and not considered further.  The U-Pb data
were used as inputs into the updated version of the discordance modeling algorithm of Reimink et
al. (2016). The results of modeling are shown in Fig. 8. As discussed previously, the discordance
modeling produces a 'sum likelihood' value across a range of lower intercept ages, in this case
from 0 to 50 Ma. Note that the lower intercept range is independent of the model outputs as the
data reduction is conducted geometrically. The most likely lower intercept age is definitively
greater than 0 Ma, as the model is capable of modeling future ages and likelihood declines rapidly
prior to 0 Ma. The peak in likelihood is ~24 Ma, and sensitivity tests (Section 3.2) show that the
best fit age is 16–28 Ma, based on the locations of the bootstrap-resampled sensitivity data (Section
511    3.2).


### 3.2 Uncertainty Analysis

In order to test the sensitivity of the results shown
in Fig. 8 to individual data points or groups of data
points and conduct uncertainty analysis, we
conducted a bootstrapped resampled modeling
routine. The full Tintic Formation dataset was
resampled with a pick-and-replace method to
create 1000 synthetic datasets that contained the
same number of data points as the original dataset
(407 analyses). Each of these synthetic datasets
was put through the discordance dating method
and summed likelihood lower intercept ages were
calculated. The maximum summed likelihood
values and the ages of those maximum values
were aggregated for each synthetic dataset. Figure
8B shows the distribution of the 1000 resampled
discordance models, the majority of which have
lower intercept peaks that cluster between 18 and
28 Ma. These ages are centered on the youngest
ages documented by Stearns et al. (2020) from
endoskarn titanites with ages down to 23 Ma
which have been found in the Alta Stock
metamorphic halo. Our data show that
discordance induced by metamorphic resetting
and/or fluid induced Pb-loss at this time affected
a large portion of detrital zircons in the Tintic
formation and validates our theoretical approach
to discordance dating of disturbances in detrital
zircons.

Non-radiogenic Pb, termed 'common lead', can
be found in meaningful concentrations in zircon,
particularly metamorphic or otherwise perturbed
zircon crystal lattices (e.g., Andersen, 2002;
Schoene, 2014). The correction of this common
Pb is necessary for accurate and precise
geochronology, and several algorithms can be
applied to remove the effect of common Pb from
the U-Pb age calculations. The Tintic formation
detrital zircon analyses do contain some common
Pb, with higher concentrations present in a
fraction of the more discordant analyses. To test if

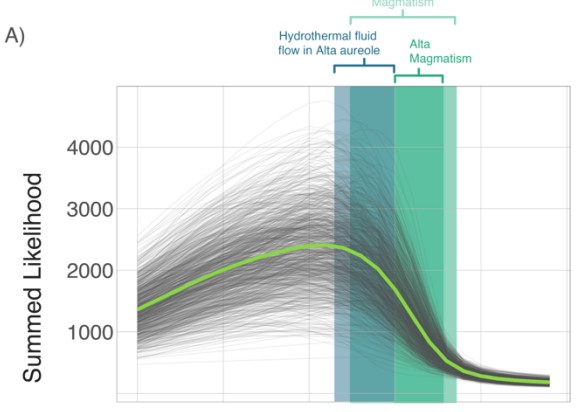

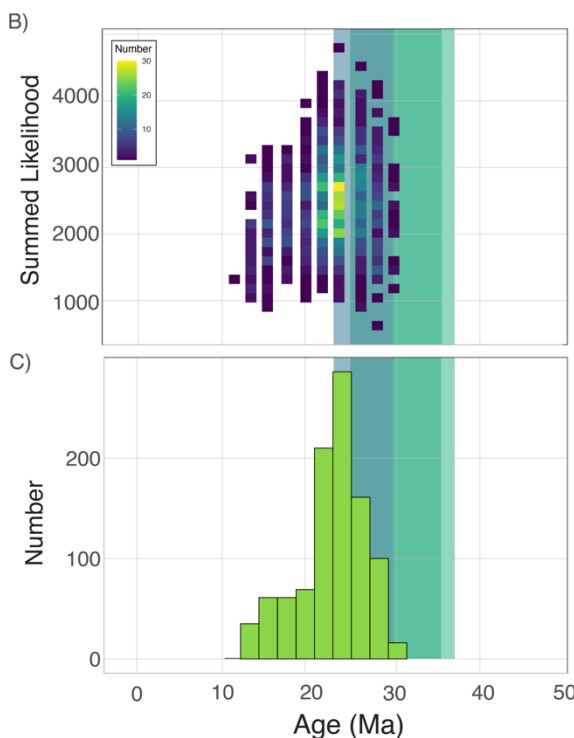

*Figure 8: Outputs from discordance dating of Tintic
formation zircons, and bootstrapped resampling efforts. A
shows the summed likelihood lower intercept results of
discordance dating. B shows the age and maximum sum
likelihood value for each maximum value calculated from
the curves shown in A. C shows a histogram of the peak
positions in Ma. The discordance dating age can be
estimated based on the resampling method to be 24 +4/-8
Ma based on the location of the median, 5th, and 95th
percentile peak locations shown in C.*

common Pb corrections influence our discordance dating methodology, we analyzed the Tintic
formation zircons after two different common Pb correction types, in addition to the uncorrected
data shown in Fig. 8 (green line in A). These results are shown in Figure 9. First, we apply various

common Pb correction methods to subtract the contribution of 'common Pb' to the [206]Pb and [207]Pb signals. One common method (e.g., Schoene, 2014) is to use the [204]Pb signal as an indication of the amount of common Pb, assume a Pb isotope composition using the Stacey-Kramers (Stacey and Kramers, 1975) global Pb-isotope evolution model, and subtract this common [206,207]Pb from the presumably radiogenic portion of the Pb. Another method involves the iterative solution to a series of equations assuming that the U/Th has not been disturbed and any time of Pb-loss is known (Andersen, 2002) – the latter assumption makes the method inappropriate for use with highly discordant data where the age of isotopic disturbance is unknown, data similar to the Tintic data evaluated here. Nevertheless, we show that there is little difference between the outputs of our discordance dating model on U-Pb data corrected using different common Pb

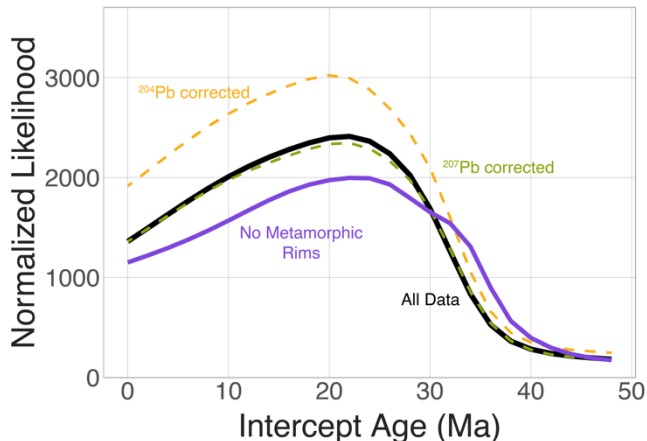

Figure 9: Plot showing the summed lower likelihood for Sample 2 U-Pb data with various common-Pb corrections applied. The black curves shows the base data with no common Pb correction. The orange curve shows the same data with a [204]Pb and Stacey-Kramers common Pb correction applied. The green curve shows the [207]Pb or Anderson correction applied to the same data. The purple line shows the lower intercept result for data that excludes all analyses that included a metamorphic rim growth feature. There is little change in the peak position across these various datasets, indicating that common Pb is unlikely to be affecting our results.

calculations (Fig. 9), suggesting that the discordance dating lower intercept age is robust from the influence of common Pb, at least for the Tintic Formation zircon dataset evaluated here.

Finally, uncertainty in the decay constants of [238]U and [235]U (Jaffey et al., 1971) may impact the discordance dating procedure to various degrees depending upon the age range of the zircon U-Pb data. Our discordance dating process fundamentally maps isotopic probability across Pb/U space, thus is not strictly dependent upon the decay constant uncertainties. Additionally, decay constant uncertainties are systematic uncertainties – meaning they apply to all data equally. In other words, if the 'true' [235]U decay constant is lower than measured by some degree, it will always be lower to that same degree. We can therefore perform a conservative assessment of the potential impact of uncertainty in the decay constants of the U isotopes by re-running the discordance dating procedure and simply changing the decay constants within uncertainty. In effect, this serves to shift the equal-age concordia line one direction or another in Pb/U isotope space, which changes the location of the age-ratio intercepts. For the Tintic formation zircon data, we can model the extreme examples of decay constant uncertainty by lowering the [238]U decay constant by 0.107% and increasing the [235]U decay constant by 0.137% (Jaffey et al., 1971), running the discordance dating procedure, then rerunning by raising the [238]U decay constant and lowering the [235]U decay constant. For the Tintic formation detrital zircon data, there is no change in the peak position of the lower intercept summed likelihood metrics, indicating that decay constant uncertainty does not impact the results of this study. However, decay constant uncertainty can affect interpretations of zircon populations that have significantly older discordance on the order of half a percent of the age. We have implemented a switch in the codebase allowing users to perform this uncertainty evaluation on their own datasets, but must note that this test may result in an unrealistically high estimate of

decay constant uncertainty as it does not take into account any correlation in the decay constant
errors that may exist (e.g., Mattinson, 2010).
At the current time, definitively distinguishing between the various types of isotopic disturbance
and resetting processes is not possible. However, differentiation could be done by, for instance,
combining zircon U-Pb-TE data with Raman spectrometer analyses of altered domains that could
quantify the crystallinity of the zircon lattice (Anderson et al., 2020; Nasdala et al., 2010; Resentini
et al., 2020; Zhang et al., 2000) and help determine which of these two models, metamorphic
recrystallization and/or fluid-induced Pb loss, was a dominant Pb-loss-inducing process in the
Tintic zircon population. However, this would need to be accomplished prior to destructive
analyses for U-Pb-TE via laser ablation. Addressing the micro-scale causes of U-Pb discordance
in detrital zircons will be important for evaluating the range of geologic settings where discordance
dating could be reliably applied to the rock record.
*3.3 Comparison with other discordance dating approaches*
The Concordant-Discordant Comparison method uses an approach that inverts discordant U-Pb
data using various lower-intercept ages, then compares the reconstructed upper intercept age
results to the concordant populations of the same sample (Kirkland et al., 2017; Morris et al., 2015)
to find the most likely lower intercept age. Here, we test the CDC method on the Tintic formation
U-Pb population and briefly discuss the outputs.
Figure 10 shows the results of the CDC method conducted on the Alta aureole Tintic detrital zircon
analyses. It is important to note that a small fraction (71 out of 407) of Tintic zircon analyses are
within 10% of concordance, so given the CDC method's reliance on a concordant fraction of data,
we should not expect this method to perform optimally. Figure 10A shows a comparison of the
CDC method to the discordance dating approach outlined in this work. The CDC method uses a
K-S statistic to compare reconstructed discordant populations with the concordant data, and the K-
S statistic returns low (near-zero) values for datasets that compare well to one another. In Fig. 10A,
we have inverted this K-S statistic such that high values (near one) represent data populations that
are more like one another. We have also normalized the discordance dating summed likelihood
metric to the maximum value (peak at 24 Ma), strictly for the purposes of comparison.

As shown in Figure 10A, the CDC method does not return peaks in the relevant statistic near 25-30 Ma for reasonable discordance cutoffs (10-20%), and only returns small, broad peaks in this time interval when the discordance cutoff is high (>30%). These results might suggest that the CDC method is highly sensitive to the fraction of concordant data present in any given sample and shows that resolving lower intercept ages without reliable concordant data, as in the case of the Tintic detrital sample analyzed here, is difficult.

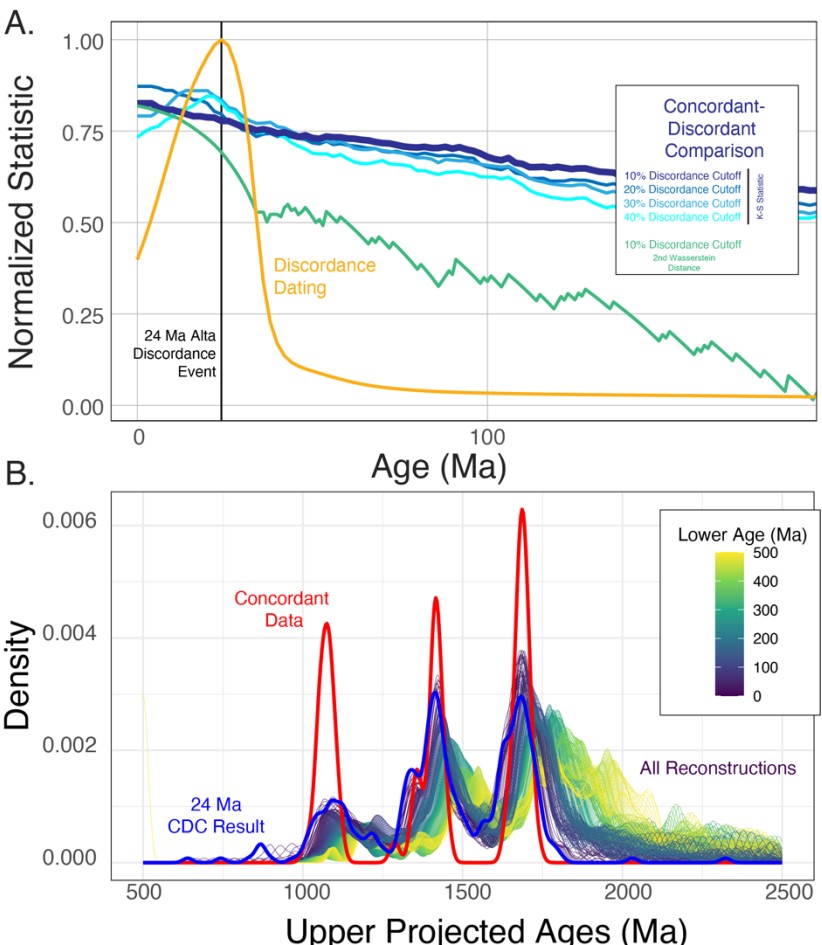

However, a dearth of concordant data may not fully explain the lack of resolvability when using the CDC method. Figure 10B shows kernel density estimators for a wide range of reconstructed upper intercepts (all using a 15 Ma bandwidth, chosen for clarity). The red line shows the age of the 10% concordant data from the Tintic formation, which is used by the CDC method as the comparator and assumed to be the 'true' distribution. The colored lines are reconstructions of the discordant data fraction, colored by the age of the lower intercept used for the upper age reconstruction. The bold blue line shows the reconstructed upper ages calculated using a 24 Ma lower intercept age. The reconstructed upper ages calculated using a 24 Ma lower intercept age do

*Figure 10: The Concordant-Discordant Comparison method applied to the Alta Tintic detrital zircon data. Panel A shows a normalized discordance dating result (normalized to the peak summed likelihood at 24 Ma), compared with the results of five different CDC model runs. Each run uses a different discordance cutoff for determining the concordant and discordant populations (10%, 20%, 30% and 40%). Note that we have inverted the K-S statistic and normalized + inverted the Wasserstein metric such that similar datasets will have a value closer to 1, again for comparison's sake. Panel B shows raw outputs from the CDC method across a wide range of ages. The red curve shows the 10% concordant data distribution. The colored curves show reconstructed upper intercept age distributions colored by the age of lower intercept used for the reconstruction. The bold blue labeled line shows the reconstruction using a 24 Ma lower intercept. Reconstructions are created using lower intercepts every 2 Ma from 0 to 500 Ma. Reconstructed peak locations match well for the 24 Ma lower intercept reconstruction, but the relative peak heights are significantly different, suggesting the K-S metric, which relies on a cumulative density plot comparison, is unlikely to give reliable comparison metrics.*

have age peaks that match the peak locations of the concordant data (red line), however as shown

in Fig. 10A, the K-S comparison statistic does not return this lower intercept age as a likely time
of discordance, except for minor, <24 Ma peaks when using very high (>30 %) discordance filters.

This suggests that part of the reason for the lack of lower intercept age resolution for the Tintic
formation detrital zircon data returned by the CDC test might be due to the comparison test statistic
itself. The K-S statistic returns the maximum vertical difference between two cumulative
distribution curves (Massey, 1951) such that 0 is a perfect match. The reliance of this statistic on
cumulative distributions means that the K-S metric is sensitive not only peak locations, but also to
relative peak heights. The dependence on relative peak heights is by design, and often implemented
in detrital zircon comparison statistics, as the relative proportion of grains of specific ages can be
of importance for detrital zircon geochronologists. Thus, most comparison statistics applied to
detrital zircon datasets are designed to also evaluate peak height differences (see the excellent
overview provided in Vermeesch, 2018b), and even the Wasserstein distance metric (Lipp and
Vermeesch, 2023) does not identify a 24 Ma peak when relying on a comparison to concordant
data filtered at the 10% level. It is possible that the use of comparison statistical tests may be
limiting the use of the CDC method for the Tintic zircon data presented here (Fig. 10B), which
experienced a discrete overprinting event ca. 30-24 Ma. For instance, the 1.1 Ga reconstructed
peaks are far lower than the 'concordant' data peak at that same time, while the 1.8 Ga
reconstructed peak is relatively high. This is likely causing the K-S statistical test (and other
similarity/dissimilarity metrics) to break down for this particular use case.

When dealing with discordance inducing events, biases may be imparted onto the discordant data
population due to various factors. We speculate that the Tintic 1.8 Ga detrital zircons, being older,
likely accumulated more radiation damage to the crystal lattice than the 1.1 Ga detrital grains.
Thus, during a 24 Ma alteration event, the 1.8 Ga population may have experienced more
disturbance, leading to more discordant data. This scenario would impart a dramatic bias in any
statistical comparison between the discordant and concordant data that relies on cumulative density
curves (e.g., K-S test, Wasserstein distance, etc.). As shown in Fig. 10B, the reconstructed upper
age distribution using a lower age of 24 Ma does indeed reproduce the peak locations well, though
not the relative peak heights. Further testing outside the scope of the present work is required to
fully evaluate the real-world applicability of various discordant data treatment approaches, where
they perform well and where they may break down.


**4. Outlook and Future Directions:**
Here we have showed that discordance dating may be used to date discordance-inducing events
that affected detrital zircon populations. We have shown that this method provides several million-
year age resolution on alteration events in rocks that experienced ~24 Ma fluid flow and
metamorphism at temperatures that reached ~450 °C. The high temperatures of metamorphism
within the Tintic formation sample may lead to the perception that discordance dating is a tool that
is applicable to high-T metamorphic settings only. However, several pieces of data suggest that
discordance dating may be an impactful method with wider applicability.

First, most zircon geochronologists pursue readily interpretable and therefore concordant data.
Geochronologists tend to select spots targeting regions of high zircon quality and pick laser or ion
probe analytical locations that are likely to retain a closed U-Pb system – a single metamorphic

growth rim or a single domain of clearly igneous zircon. Additionally, what little discordant data may have been collected in laboratories around the world is commonly filtered based on a discordance threshold. Thus, there is no clear way to determine the general prevalence of discordant data in detrital zircon datasets. Additional analytical focus on discordant grain volumes may lead to additional insight into the value of discordant data for geochronological purposes.

Second, discordance dating will theoretically become more precise if discordant grains have older initial crystallization ages and the resetting event is relatively young. Due to the geometry of U-Pb isotope space, older grains experiencing more recent discordance-inducing geological events will provide a more precise estimate of lower intercept ages. However, though the Tintic formation contains grains with primary ages up to ~1.8 Ga, other more ancient grains would provide additional precision to any discordance dating analysis. The larger the age dispersion in detrital zircon crystallization ages, the more precision discordance dating would yield for Phanerozoic alteration events. Thus, a sediment that experienced less aggressive fluid flow or metamorphism, but that had older detrital zircon grains, may yield important lower intercept age information, with uncertainties on the order of a few million years.

Third, there is a growing body of evidence that discordance in zircon can be induced in a wide variety of low-temperature environments. Both experimental (Geisler et al., 2001; Pidgeon et al., 1966) and empirical (Geisler et al., 2002; Morris et al., 2015; Pidgeon et al., 2016; Zi et al., 2022) studies have documented large degrees, and sometimes near total, discordance at much lower temperatures (100–200 C) than the Tintic formation zircons analyzed in this work experienced. Thus, it is possible that many suites of detrital zircons experienced low-T hydrothermal fluid flow events that induced Pb-loss and/or recrystallization to such a degree that discordance dating could provide useful age information.

Such information could prove useful when paired with other low-temperature chronometers dating such events as diagenetic xenotime growth (McNaughton et al., 1999), zircon cooling below 200-400 °C using Raman dating (Härtel et al., 2021), and U-Pb/U-Th-He double dating to derive low-T geological histories of individual grains. It is important to note that Raman dating and U-Th-He dating of zircon are both temperature sensitive phenomenon that would ideally target pristine zircon growth zones, whereas discordance dating requires analyses of damaged, discordant grains. The result is that these various techniques could be applied to the same grains within a given population to both extract detailed thermo-chemical histories, as well as intercalibrate the various dating techniques. Notably, more work is required to robustly determine the sensitivity of discordance dating to both fluid alteration and elevated temperatures. Such information will provide insight into the interplay between different low-T geochronometers applied to detrital zircons. Discordance dating may be particularly interesting as there are a limited number of geochronometers that are currently available to date reactive fluid flow events (brine migration, mineralizing fluid flow, low-T metamorphism, etc.) in sedimentary rocks– even imprecise age information is useful in such scenarios.

**5. Conclusions:**

We have documented a new tool for geochronologists to date in situ detrital zircon discordance-inducing events. We have shown the utility of this technique using synthetic datasets and ground-truthing by analyzing zircons within the well-studied Alta Stock metamorphic aureole. The discordance dating technique returns discordance ages of ~24 Ma, which is the expected age of fluid flow in this region. Our method may have significant applications to determining rates and absolute dates of diverse geologic phenomena; basin brine migration, mineralizing fluid flow, and low-grade burial metamorphism are just a few of the processes that may induce discordance in zircon analytical volumes such that it is amenable to discordance dating.

**Acknowledgements:**
This manuscript was substantially improved by reviews from Ryan Ickert, Axel Schmitt, and Chris Kirkland, as well as editorial handing by Daniela Rubatto. This work was supported by Rudy L. Slingerland Early Career Professor of Geoscience funds awarded to Jesse Reimink, and PSU Geoscience Department undergraduate research funds awarded Renan Beckman.

**Data Availability:**
The data reduction code and raw data used for discordance modeling is available here:
https://github.com/jreimink-isotope-geochem/discordance-dating and at the following Zenodo DOI 10.5281/zenodo.13972610
and a public-facing easy-to-use Shiny app is available here:
https://discordance.geosc.psu.edu/discordance_app/

**Competing Interests:**
The authors declare that they have no conflict of interest.

**Author Contributions:**
JR, JD, and ML conceived the study. AS carried out sample collection. RB, ES, AC, and JG separated zircons, collected U-Pb-TE data, and produced final results. JR and JD carried out modeling and sensitivity testing. All authors contributed to final data interpretation and manuscript production.

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
