# Peer review of "Discordance Dating: A New Approach for Dating Alteration Events"

_Geochronology, 2024_

## Author Comment (AC1)

This contribution is about the potential of detrital zircon U-Pb dates to record geologic events that overprinted zircon and caused partial to near-complete Pb-loss, resulting in discordance. Such discordant detrital zircon data are usually discarded, whereas Reimink et al. convincingly argue that such data can reveal geologically meaningful lower concordia intercept ages. A numerical model is introduced, building on Reimink et al. (2016), and tested against a real-world data set for detrital zircon from a quartzite which was thermally overprinted in a magmatic contact aureole.

Judging from the success of Reimink et al. (2016), which was designed to extract meaningful formation ages from discordant detrital zircon data sets and is frequently cited, I find this approach promising. Interrogating detrital zircon for their potential of revealing geological episodes capable of producing Pb-loss would circumvent culling of significant amounts of data and allow gaining useful insights from zircon domains usually not targeted for high spatial resolution analysis due to their non-ideal structure (e.g., rims, cracks, etc.). The quality of the writing and the artwork are at a high level, and the scope of the work is a perfect match for GEOCHRONOLOGY.

**We thank Dr. Schmitt for his robust and constructive comments. They have helped us significantly improve the manuscript.**

One suggestion for improvement is adding an explanation about the lower age limit of this approach. The manuscript clearly states that older primary ages permit more precise identification of discordance, but it does not specify the lower age limits. Discordance in Phanerozoic detrital zircon ages is typically difficult to discern in LA-ICP-MS or SIMS data, and $^{207}Pb/^{206}Pb$ ages required as input in the model will have high uncertainties. I tested the model with the link provided on the Ulusoy et al. (2019) dataset, and it did not produce the expected zero-age intercept for heating during a Holocene eruption.

**This is an excellent point. To address this we have conducted an additional set of sensitivity modeling and added one additional figure. This section builds on the previous modeling work except allows the maximum age of any zircon in a synthetic population to decrease to 100 Ma, all with a 30 Ma imposed discordance-inducing event. This allows us to quantify the necessary ages of zircon grains that would provide sufficient precision to any given lower intercept age calculation. This comment led us to also model an increase in maximum zircon age, well into the Archean, such that we can provide guidance on the precision capabilities of this method as a function of the primary ages of zircons in the sample database.**

**The results of this modeling were surprising to us, in that older grains to not universally provide more precise discordance dating ages. We have added additional discussion to this section to fully address the results of this modeling, including providing guidance on sample selection.**

**With regard to the Ulusoy et al dataset, the reason for the lack of a zero age intercept is straightforward. There are only a few >1 Ga discordant data points in that dataset, and several near-concordant Phanerozoic grains. This means that the older discordant grains provide likelihood along chords that intersect with many of the younger grains, producing a flat likelihood structure across much of the Phanerozoic. We have provided some discussion of the types of data necessary for reliable use of the discordance dating technique, and discussed some analytical approaches that may prove useful to interested geochronologists.**

Potential mechanisms which produce Pb-loss in zircon are discussed. The authors specifically address recrystallization/overgrowth vs. fluid induced leaching for their sample data considering correlative trace element data. Although they are correct that identifying the Pb-loss mechanism is difficult and may require different tools on a case-by-case basis, it would be helpful to provide a bit more context by discussing other zircon-based dating methods targeting sediment evolution (e.g., in-situ dating of xenotime overgrowths, U-Th/He geochronology, Raman dating, or fission tracks; see additional reference list).

**This is an excellent suggestion. We have added some discussion of these techniques and the suggested references to the main text.**

If possible, it would be useful to quantify the thermal regime for which this new discordia-lower-intercept geochronometer/thermochronometer is sensitive, for example by calculating model closure temperatures for volume diffusion (e.g., for Pb in metamict zircon; Geisler et al., 2002) and comparing these to those of alternative methods mentioned above.

**This suggestion is an interesting one. After careful consideration we have decided not to include a volume-diffusion model in this manuscript. Given the correlation in trace-element concentrations with discordance in the Tintic detrital zircons, pure diffusion of Pb is unlikely to be a dominant discordance-inducing mechanism in our sample set.**

In the list of processes suspected of causing discordance (Lines 69–76), I would also include pyrometamorphic heating for completeness. Zircon in crustal volcanic xenoliths or contact rocks when sufficiently heated can also be (partially) reset; this has been

utilized by (U-Th)/He dating (Cooper et al., 2011), and concomitant Pb-loss has also been documented (Ulusoy et al. 2019).

**Excellent suggestion, changed as recommended.**

Providing an easy-to-use portal for the numerical model is a welcome service to the community. When testing it, however, I missed an output value for the lower intercept age and its uncertainties.

**We have not implemented a single procedure for outputting lower intercept age and associated uncertainty as the outputs from any given sample's discordance dating procedure will be widely varying. The output data can be downloaded and a knowledgeable chronologist who is intimately familiar with the vagaries of their particular dataset can apply peak fitting and uncertainty estimates (FWHM, etc) to that data. We do note that some datasets would be inappropriate to input to this type of data modeling procedure, and outputs such as suggested here could potentially be misleading. Thus, we prefer to provide output data and leave the interpretation, and especially uncertainty assessment, to the individual chronologist.**

Some additional suggestions for improvement and minor corrections are provided point-by-point.

Line 32: Please write "U-Th-Pb" as the Schaltegger et al. (2015) also reviews U-Th disequilibrium dating.

**Changed as suggested.**

Line 35: Please check references for completeness; none of the three references cited here were found in the reference list.

**Thank you for catching this error, it has been corrected.**

Line 64: Micron = not SI; should be micrometre

**Corrected**

Line 75: Pyrometamorphic heating of xenocrysts/xenoliths is another process (Ulusoy et al., 2019).

**Reference added**

Line 98: It would be helpful to explicitly state the formula for calculating discordance here, as it was done in Reimink et al. (2016).

**Changed as suggested.**

Line 114: The discordance method can be seen as complementary to (U-Th)/He dating or other methods in its ability to extract thermally or fluid induced alteration of sediments. Mentioning these alternative approaches would provide valuable context.

Line 150: Something is missing here.

**"are the most useful" was added here to complete the sentence.**

Line 165: Here and elsewhere: ranges should be indicated by the "en dash".

**Changed as suggested.**

Line 192: "and" after 1800 Ma?

**Changed as suggested**

Line 150: Why 150? Please justify.

**We have added a line justifying this number of samples.**

Line 277: Isn't this a logical consequence of each probability curve being normalized to an area of unity?

**Yes and no. The dataset is internally normalized to the probability distribution of a given number of datapoints in a dataset. Given that all the simulations here have the same number of data points, the total area under the curves will be fixed when considering all of the U-Pb data space (0-4500 Ma, including Discordia arrays). However, we are focusing on a small area of U-Pb space here, such that the area under any of the curves in Fig. 3 (C, F, I) are not identical. The peak height is much more strongly correlated to the position of the most discordant data point, which serves to focus likelihood to a single point along a lower intercept array (both increasing peak height and sharpening the peak).  We have added a line explaining this in the text. "Note that in this version of the analysis, we are focusing on a small portion of the total U-Pb Discordia space, such that the area under each individual curve in Fig. 3C, F, and I are not all uniform."**

Line 354: Space between number and unit.

**Changed as suggested**

Line 398: Use official name SRM 612
([https://tsapps.nist.gov/srmext/certificates/612.pdf](https://tsapps.nist.gov/srmext/certificates/612.pdf))

**Changed as suggested**

Line 398: When comparing data for the 91500 secondary reference zircon to literature values, some discrepancies are noted. Campbell et al. (2014), for example, state 11 +- 3 µg/g Al in 91500 (1se), whereas the average from the supplement is only half that value (5.7 +- 0.19 µg/g Al). Notably, there is also significant scatter in the data (MSWD = 5.1). The discrepancy is even more severe for Ca, for which literature values are 1.9 +- 0.6 µg/g (Coble et al., 2018) whereas the average for the data in the supplement is 35 µg/g (with in part very large uncertainties and even negative values). Iron in 91500 zircon, by contrast, is lower in the supplementary data compared to the literature (1.71 vs. 3.4 µg/g; Coble et al., 2018). I am suspicious about these elements being major components in NIST SRM 612 glass (except for Fe): Al and Ca are present at ~2 and ~12 wt.% (oxide) levels. How much of a matrix effect does this introduce when NIST SRM 612 is used as the trace element primary reference material for zircon? If trace element data are inaccurate for zircon, then raw ratios should be used, which would serve the same purpose. Please also remove negative values from the supplementary table and state corresponding detection limits.

**Excellent questions, though much of this text will be removed based on suggestions by Dr. Ickert. However, here are some thoughts. First, 91500 has been shown to be heterogeneous in trace element composition (Caulfield et al., 2025, Chemical Geology), specifically Al which has a wide range (0-15 ppm) of concentrations. This is shown below where the grey box is the range in Al in 91500 documented by Caufield et al. (2025).**

[Figure]

Conversely, our GJ-1 data shows clearly that the concentration of Al is accurately reproduced using NIST SRM 612 as the primary standard (mean of 3.8 compared to the accepted value of 3.75, see below). This gives us confidence that at least to a first order our trace element data using NIST SRM 612 as a primary reference material is working properly.

[Figure]

We agree with Dr. Schmitt that the accuracy of the Ca and Fe concentrations are hard to evaluate, largely due to a lack of information on zircon reference materials. However, we have chosen to keep the concentration data while adding in the raw ratios to the final reference material data table. We have kept the concentration data because the high calculated concentrations are, at least in a rough sense, indicative of the changing chemistry of altered zircon domains. We believe presenting trace element data as concentrations will be useful to the zircon geochemistry community and might help guide further work on zircon alteration mechanisms, including those causing U-Th-Pb discordance.

**We also note that in our Session #3, there is excess scatter in the Ca, Fe, and other low concentration elements. This was due to a background issue during this run that affected some of the low concentration elements, yielding many negative values in the resulting concentrations. However, this does not affect high concentration elements (e.g., Yb, Hf) and does not seem to dramatically affect the interpretations of our sample data due to the high concentrations of Ca and Fe detected in the Tintic detrital zircons, as shown below.**

[Figure]

Line 401: Please address why the $^{207}Pb/^{206}Pb$ values for NIST SRM 612 appear to be significantly lower than reference values reported in the literature (0.8995 vs. 0.907; Woodhead and Hergt, 2001)? Also, there are several outliers for run IDs between 500 and 531. How does this affect the robustness of the zircon $^{207}Pb/^{206}Pb$ results analysed under these conditions?

**This difference is minor (~0.7% lower) and the NIST SRM 612 $^{207}Pb/^{206}Pb$ ratios are generally within uncertainty of the accepted value, despite the different matrix composition (shown below).**

[Figure]

The beginning analyses of Run #3 do fall significantly outside of uncertainty of the accepted value. But these do not affect the samples or standards analyzed later during this session, and do not impact the trace-element data collected on the iCAP-RQ mass spectrometer. We have added text to the manuscript, and additional supplemental figures in the GitHub repo documenting this and further U-Pb data descriptions (reference in a revised version of the manuscript).

Line 403: Spelling: Peixe (here and elsewhere)

**Changed**

Line 449: between … and

**Changed as suggested**

Line 464: In Fig. 7, please state a value and an uncertainty for the discordance date.

**We have chosen to use the median, 5th, and 95th percentile peak positions derived in the bootstrapped resampling method to determine the uncertainty in this age. This is now reported in the caption of Fig. 7 and changed throughout the manuscript.**

Line 523: Fig. 9 preferable μg/g instead of ppm (cosmetics: superscript in panel C).

**This text is removed following Dr. Ickert's comments.**

Line 526: Al-in-zircon as a tracer for discordance is interesting, and a bit surprising as Al is comparatively fluid immobile. The dissolution-repreciptation scenario for metamict zircon invokes amorphous phases in recrystallized zircon as sinks not only for Al, but also Ca and Fe (e.g., Geisler et al., 2007). It is hence unexpected that Ca and Fe seemingly do not share the trend for Al. In the light of the deviations of the reported values for secondary references from literature values (see comment for line 398), could you please comment if such variability could have gone undetected?

**This text would be removed following Dr. Ickert's comments. However, there is a weak correlation between Fe and Al, and Ca and Al, when considering the sample data plotted in log-log space (shown below). Though far from a perfect correlation, particularly for Ca, there a general consistency of behavior between these elements, especially for Fe. This may indicate that the mechanisms suggested by Geisler et al. 2007 were operating in Alta zircons, though we are not confident enough in that assessment to substantially modify our interpretations.**

[Figure]

Line 547: Please explain how alpha dose was calculated.

**This text would be removed following Dr. Ickert's comments.**

Line 582: The first column is difficult to understand; can the percentiles be separated from the classes, and be directly shown with their respective columns?

**This text would be removed following Dr. Ickert's comments.**

Line 632: Please add degree symbol. This would also be the place to discuss the thermal sensitivity ("closure temperature") of different chronometers applicable to zircon.

**This text would be removed following Dr. Ickert's comments.**

Line 641: "to use" seems superfluous

**Removed as suggested**

Line 668: Please use abbreviations that are consistent with the author list.

**Changed as suggested**

Additional references

Campbell, L. S., Compston, W., Sircombe, K. N., & Wilkinson, C. C. (2014). Zircon from the East Orebody of the Bayan Obo Fe–Nb–REE deposit, China, and SHRIMP ages for carbonatite-related magmatism and REE mineralization events. Contributions to Mineralogy and Petrology, 168, 1-23.

Coble, M. A., Vazquez, J. A., Barth, A. P., Wooden, J., Burns, D., Kylander-Clark, A., ... & Vennari, C. E. (2018). Trace element characterisation of MAD-559 zircon reference material for ion microprobe analysis. Geostandards and Geoanalytical Research, 42(4), 481-497.

Geisler, T., Schaltegger, U., & Tomaschek, F. (2007). Re-equilibration of zircon in aqueous fluids and melts. Elements, 3(1), 43-50.

Geisler, T., Ulonska, M., Schleicher, H., Pidgeon, R. T., & van Bronswijk, W. (2001). Leaching and differential recrystallization of metamict zircon under experimental hydrothermal conditions. Contributions to Mineralogy and Petrology, 141(1), 53-65.

McNaughton, N. J., Rasmussen, B., & Fletcher, I. R. (1999). SHRIMP uranium-lead dating of diagenetic xenotime in siliciclastic sedimentary rocks. Science, 285(5424), 78-80.

Reiners, P. W., Campbell, I. H., Nicolescu, S., Allen, C. M., Hourigan, J. K., Garver, J. I., ... & Cowan, D. S. (2005). (U-Th)/(He-Pb) double dating of detrital zircons. American Journal of Science, 305(4), 259-311.

Woodhead, J. D., & Hergt, J. M. (2001). Strontium, neodymium and lead isotope analyses of NIST glass certified reference materials: SRM 610, 612, 614. Geostandards Newsletter, 25(2-3), 261-266.

Ulusoy, I., Sarıkaya, M. A., Schmitt, A. K., Şen, E., Danišík, M., & Gümüş, E. (2019). Volcanic eruption eye-witnessed and recorded by prehistoric humans. Quaternary Science Reviews, 212, 187-198.

---

## Author Comment (AC3)

**We thank Dr. Ickert for the careful and substantive comments. These points have helped substantially improve the manuscript.**

In this manuscript, the authors do the following:

- Present a modification (section 2.1) to an algorithm introduced in a prior publication (Reimink et al., 2016) so that "lower intercept U-Pb concordia" data can be inverted from discordant sets of U-Pb measurements. They ground-truth the algorithm using a suite of synthetic datasets (section 2.2).
- They apply this algorithm (section 3) to a suite of detrital zircon data from a Cambrian clastic rock affected by a well constrained Oligocene-Miocene (~25-23 Ma) magmatic and hydrothermal event. Inverting the highly discordant array of zircon U-Pb data recovers a ca. 25 Ma date.
- They describe trace element and imaging data collected on the zircon (section 3.3), and speculate on its utility for inferring discordance inducing processes.

The manuscript is well-written, clear, contains high quality geochemical and isotopic data, and has good figures, although in the PDF the figures are quite small and difficult to read.  I assume that in a final typeset form this would be rectified.  The modification to the 2016 manuscript should be useful.

Since the manuscript is so generally well written, I only have large-scale "substantive" comments.

***Manuscript length***

The manuscript is too lengthy and should be a "short communication" or a "technical note".  The heart of it is a modest, straightforward (but useful!) modification of a technique introduced in a 2016 paper.  The introduction is a suite of boilerplate "zircon is a good mineral" text, and the entirety of Section 3.3 has no bearing on the conclusions drawn in Section 3.1/3.2 and is not referenced in the abstract.  The trace element/imaging data in Section 3.3 is interesting and the authors have obviously put a lot of thought into it but unfortunately -as they make clear – the results provide no real insights.  If the authors would like to infer process from trace element data from discordant zircon, I would like to read that, but it should be a different manuscript.  I strongly recommend cutting section 3.3 (the rest of the manuscript would read the same with no editing, including the abstract, introduction, and conclusion), and compressing the introductory material (it's just a list with no literature synthesis) and if needed, moving some or all of section 2.2 and 3.2 to a supplement.

**This is a worthwhile suggestion to consider. We appreciate that the trace element discussion (formerly 3.2) is not conclusive and have removed that discussion. We have also reduced the length of the discussion of sensitivity modeling, though not completely removed it as this discussion highlights important interpretative limits on the discordance dating tool presented in this work.**

**As far as transitioning the manuscript to a Technical Note, we would prefer to not make this change. The guidance for the length of a Technical Note in GChron is provided as "a few pages only", which would require us to cut most of the text in this manuscript, including the sections on sensitivity testing as well as prevent us from adding in new discussion requested by comments made by all three reviewers. In our view, the modeling and sensitivity testing is a valuable component to our efforts as it shows the statistical limits of the method proposed here. We would, therefore, be hesitant to remove all the text and figures that discusses the limits of our method as required by the length of a Technical Note.**

**As an alternative, we have kept this manuscript a research article, while removing Section 3.3 (trace element discussion) and shortening the introduction (the specifics of which are outlined below in response to a comment by Dr. Kirkland), and the text of Section 2.2. This has removed much of the text highlighted as unnecessary by Drs. Kirkland and Ickert, while keeping the text that is germane to the discordance dating procedure specifically. This also allowed us to add additional text required to address some of the points made by Dr. Schmitt (regarding uncertainty) and Dr. Kirkland (regarding other methods of discordant data treatment).**

**We believe that including the modeling discussion, while still having several figures to fully visualize the value of discordance dating makes the manuscript more readable for the user. However, we would ultimately defer to the editor's recommendation on this decision.**

***Singular discordance events***

The passage on line 108-113 describes a critical assumption for this technique:

"…one useful assumption can safely be applied: after the deposition of the sediment, all the zircon grains have a shared thermal and geological history. In this study we leverage this assumption that post depositional U-Pb isotopic discordance may affect all zircon grains within a given sediment at the same time, in order to use discordant detrital zircon U-Pb data to investigate post-depositional geologic events."

This is clearly a safe assumption for the Alta example here, where there is overwhelming geological, geochemical, and geochronological evidence for a massive ca. 25 Ma event. It's unclear to me that this might be equally true for sample suites with different histories, including and especially those without such a strong, singular event. The key assumption here is that each individual measured chronometer responds in the same way to the shared geological history, and it's one that I suspect is not correct. Individual grains, particularly detrital grains, will have different sizes, alpha-parent concentrations, alpha-dose histories, and annealing histories and will each have different susceptibility to geological events.

**We may not understand this comment precisely, but we do not assume that each grain responds to the shared geological history the same way. In fact, the variable response is what allows for discordance dating to be useful. Our phrasing in this section was indeed confusing and did not clarify our understanding correctly, thus we can easily modify this text to add the clause "However, importantly, each grain will respond to these geologic events differently due to the unique crystallization, radiation damage, and previous thermal/annealing history of that particular grain. This shared history, but variable response, can be utilized to estimate ages of lower-temperature events than are typically recorded in zircon U-Pb ages"**

For example, fluid flow is likely to be highly protracted, and different grains are likely to respond differently, or not at all, in a manner corresponding to their local environment and history. One grain might record an event at one point because it is associated with a vein and fluid flow, then it might seal, and millions (or 10s of millions etc.) of years later a different event occurs to a different grain. Protracted uranium uptake is well documented in for example, in the literature of U-daughter product geochronology of low temperature phosphate and carbonates (a good example is some of the U-Pb data in the supplement to Fassett et al. 2011 https://doi.org/10.1130/G31466.1).

This is not to say this isn't a useful technique, but the authors are presenting, in my opinion, an inadvertently misleading characterization of the applicability of the assumption listed on lines 108-113.

**We think that the correction mentioned above would adequately address this comment by Dr. Ickert, but this latter statement is also true – discordance dating is not likely to be as precise of a tool as typical U-Pb zircon geochronology due to both the varied response as well uncertainties in the likelihood distribution (as modeled in Section 2.2 and now 2.3).**

***Decay Constants***

There is a subtle but important issue here, having to do with decay-constant uncertainties.

When single decay constants are used, and used in the same manner (for example comparing two 206/238 dates from concordant analyses) the decay constant uncertainties are very highly correlated and are typically negligible. This is the basis for ignoring such so-called "systematic" uncertainties when comparing dates from the same isotopic system. However, when mixing decay constants, and using them in what are effectively different proportions, they can no longer be neglected, and when looking at concordia "chords", can be surprisingly large.

To frame the issue differently, if you compare a 206/238 date to another 206/238 date, you almost certainly can neglect decay-constant uncertainties on the difference between the two dates. If you compare a 207/235 date to a 206/238 date, you cannot neglect them. If you compare a 206/238 to a 207/206 date, you cannot neglect the decay constant uncertainties, but they are not independent because they both use the 238U decay rate. Upper and lower intercept concordia dates each have a unique "mixture" of both decay rates and so cannot be neglected except when comparing them with very similar (e.g., subparallel) chords.

In the dataset presented here, because of the young age of the lower intercept and the old age of most of the grains, the decay constant uncertainties are negligible. But since this is meant to be a useful technique for future work, this may actually matter a great deal, particularly with early paleozoic and older, lower intercepts, where decay constant uncertainties when compared to say, 206/238 dates, can be 10s of Ma.

**This is an excellent point, and one we had not originally modeled the implications of. To clarify this point, we plot below various 'concordia' lines in 207Pb/235U vs 206Pb/238U space (Wetherill concordia). The black curve is the concordia line position using the mean decay constants while colored curves move the decay constants to the extremes of the uncertainty in the decay constant values (0.137% in the 235U decay constant, and 0.107% uncertainty in the 238U decay constant; Jaffey et al., 1971). Note that these uncertainties likely overrepresent the problem because they ignore any empirical refinement of the relationship between the decay constants (e.g., Schoene et al., 2006; Mattinson et al., 2010) The dots represent 1000 Ma, 1010Ma, 1020Ma, 1030Ma, and 1040Ma.**

[Figure]

**We plot the range of variability that could be produced by systematic uncertainty in the decay constants. There are four additional lines plotted, where we varied the U235 and U238 decay constants by their two sigma error estimates, and recalculated the position of the equal-age line (Concordia) given the isotope ratios measured. Covariation in these values (i.e., high U235 lambda + high U238 lambda) produce very little change in the position of the concordia line in Wetherill concordia space. However, anti-variance, where we calculate the equal-age line with a U235 lambda value 2sigma low, and a U238 lambda value 2sigma high (or vice versa) produces concordia lines that are significantly different from the mean line – blue and red curves and dots in the above image.**

Geometrically, it's easy to see when this will matter – as the slope of the chord near the intercept (lower or upper) becomes more parallel, the date will "smear" more within the uncertainty band around the concordia.  Having folks use this tool without a method to address this potentially significant source of uncertainty would be dangerous.

Unfortunately, it can be a bit complicated to address because it depends on the date you want to compare it to.  The decay constant uncertainty in the difference between a lower intercept and a 206/238 date is different than when comparing it to a 207/235 date (or a 207/206 date, or an intercept with a different slope etc.).  However, the calculation is straightforward to do numerically and could easily be incorporated into the code.

To address this issue in the most conservative way, we have modified our code to allow users to rerun the discordance dating model using anti-correlated decay constant values – in effect shifting the concordia line to the blue or red lines shown above, and re-running the modeling procedure.

As our model fundamentally operates using the Pb/U ratios and maps out probability distributions in ratio space, changing the decay constants requires simply using a different model concordia curve.
To highlight the potential effect of these systematic uncertainties, we re-ran the modeling on the Tintic detrital zircon dataset discussed in this manuscript. The lower intercept likelihood results are shown below. There is no significant (within uncertainty) change in the peak position (no change within the 1 Ma node spacing used here), and little change in the slope of the curve across the lower intercept likelihood space.

[Figure]

As noted by Dr. Ickert, the magnitude of this change will vary across geologic time, depending on the vagaries of a particular discordant U-Pb dataset. To highlight this, and document the potential effect, we generated a synthetic dataset with upper intercepts ranging from 2.5 Ga to 4.0 Ga, and induced discordance at 2.0 Ga.

[Figure]

We then re-ran the discordance modeling procedure three times using the range of possible concordia line positions (by changing the decay constants) to show the range of possible variation in discordance dating outputs produced solely by varying the decay constants. The outputs are shown below, and as expected there is more variation given the relatively large uncertainty in the U235 decay constant, and the relative importance of the 235U-207Pb system further back in time. However, the discordance dating peak shifts by 14 Ma (0.7% of the age). We again note that this is likely an overestimate of the shift as this simplistic approach to evaluating decay constant uncertainty does not account for

**empirical recalibrations of the decay constants.**

[Figure]

We have modified out code to allow users to perform this same uncertainty analysis and provided instructions on how to accomplish this. We have also added in a paragraph into the text describing the above uncertainty estimate.

---

## Author Comment (AC4)

**We thank Dr. Kirkland for reviewing this manuscript, and his comments will help us substantially clarify the manuscript.**

This study continues the concept of extracting age information from discordant zircon U-Pb data, particularly in detrital zircon suites. Traditional geochronology often discards discordant data, but this approach leverages discordance to date post-depositional geological events like fluid alteration and contact metamorphism. The paper applies a technique introduced by Reimink in 2016. The method is validated using synthetic data and applied to zircons from the Alta Stock metamorphic aureole, successfully dating a ~23 Ma alteration event. This technique should have implications for dating fluid flow, low-temperature metamorphism, and sedimentary basin evolution.

Introduction & Framing

The study builds upon Reimink et al. (2016) by introducing a useful refinement to an existing core technique. As such, it may be more effectively presented as a technical note that directly highlights the specific methodological improvement. Specifically, I am not convinced that the introduction needs to be structured with the basic conceptual framework of U-Pb geochronology and its uses. I would have thought that most / all readers of this journal would be well informed of the basic concepts. Also, there are elements of similarity with some previous works on the subject area starting the work in this way. In any case, I think it would be more efficient and productive for this work to be more direct about inverting discordant data to better understand its effect. So, in short, the work could commence around line 78 without any determent to the new insight the work aims to convey.

**This comment is similar to that of Dr. Ickert and it will be relatively simple to modify the manuscript to accommodate these suggestions. We have removed the first several paragraphs of the basic conceptual framework material, start the introduction with a discussion of discordance in U-Pb data, and treatments of such data (see below). We have expanded the already present discussion of discordant data treatments, as highlighted in specific responses below.**

How the Study Represents the Field

I am concerned that the general depiction of the field, as not aiming to use discordant data, is not an accurate portrayal of the current community knowledge. Please let me elaborate on this: U-Pb discordance has long been know and modelled via discordia regressions and their lower intercepts interpreted with various success as the times of meaningful geological events. Clearly with additional scatter from a single Pb loss line interpretation of the timing of radiogenic Pb mobility becomes difficult if not

impossible to determine using conventional regression approaches. However, as I am sure the authors are well-aware there is now a wide range of works that have proposed methods to address this complexity and extract meaningful times for Pb loss. Specifically, the following works all introduce methods to invert discordant data to resolve Pb loss times.

Sharman, G. R., & Malkowski, M. A. (2024). Modeling apparent Pb loss in zircon U–Pb geochronology. Geochronology, 6(1), 37-51. https://doi.org/10.5194/gchron-6-37-2024

**This paper was not initially cited because the modeling approach outlined in that work was designed to identify Pb-loss from a single population of magmatic grains, with the goal of better resolving igneous or volcanic ages. That work did not fundamentally use discordant data in the way other work we cited did. However, Dr. Kirkland makes a good point that the Sharman and Malkowski work deals with new approaches to discordant data, and citation in our manuscript is therefore useful to readers. We have included a discussion and citation in the revised introduction.**

Morris, G. A., Kirkland, C. L., & Pease, V. (2015). Orogenic paleofluid flow recorded by discordant detrital zircons in the Caledonian foreland basin of northern Greenland. Lithosphere, 7(2), 138-143. https://doi.org/10.1130/L420.1

**The Morris, et al., 2015 manuscript is cited in the manuscript in the section where discordant data treatment approaches is outlined. By our reading, this was the original derivation of the method that eventually came to be called Concordant-Discordant Comparison and was cited for that reason. We have developed a more robust introduction and description of this, and related, methods in the newly revised introduction, following the earlier comments by Dr. Kirkland and Dr. Ickert.**

Kirkland, C. L., Abello, F., Danišík, M., Gardiner, N. J., Spencer, C., & (2017). Mapping temporal and spatial patterns of zircon U-Pb disturbance: A Yilgarn Craton case study. Gondwana Research, 52, 39-47. https://doi.org/10.1016/j.gr.2017.08.004 747

**We had not initially cited the Kirkland et al., 2017 work as it derives the fundamental methodology from Morris, et al., 2015. However, it does appear to be the first time that the CDC method is introduced by name, so this paper is now referenced in the newly revised introductory text, which is outlined below.**

Kirkland, C. L., Johnson, T. E., Kinny, P. D., Kapitany, T., & (2020). Modelling U-Pb discordance in the Acasta Gneiss: Implications for fluid–rock interaction in Earth's oldest dated crust. Gondwana Research, 77, 223-237. https://doi.org/10.1016/j.gr.2019.07.017

**This manuscript was not cited as it did not appear to advance the methodology, but applied the existing discordant zircon treatment methods to a different sample location.**

My point simply is that a lot of emphasis is being placed on the term "most" in the statement "most modern U-Pb studies aim to minimize its effect rather than understand or use it".

**We apologize that our wording in this sentence has led to a mischaracterization of the manuscript. Our intention was to highlight the relative lack of focus on discordant zircon U-Pb data (several papers in the past several years) as compared to the broader uses of zircon U-Pb data (several thousand papers in the same timeframe), which we believe was appropriate.**

**Nevertheless, we do not wish to give the impression that zero work has gone into evaluating discordant zircon U-Pb data (work which is highlighted later in the manuscript). The changes outlined for the introductory text will hopefully alleviate these concerns.**

Nothing would be lost from the advance this work makes by better framing it in the context of the existing field working on exactly the problem addressed in this paper.

**We have now added in a full discussion of the approaches outlined above, including a discussion of the CDC method's performance on the Alta DZ data presented here.**

So, it would seem reasonable to also acknowledge there are a range of other techniques which also aim to invert discordant data to arrive at the most likely time of radiogenic-Pb mobility. Providing this context would better frame the advance of this work.

**This comment has been addressed by restructuring the manuscript to begin the introduction with a discussion of discordance and provide more detail about the various approaches that different methodologies have used to extract age information from discordant data.**

**We now start the manuscript with a short introduction to discordant U-Pb data, then outline various approaches to treating discordance, using one paragraph for each general class of models – traditional approaches, sedimentary zircon discordance events, the CDC test specifically, resolving age mixtures, and the quantified removal of discordance (Pb-loss).**

**This then flows directly into the model presented in the present work, which concludes the shortened introduction while also giving a more thorough overview of the various approaches to discordance (or Pb-loss as it is called by other works).**

Methodological comments

I am not convinced that the statement in line 96 is accurate "Without the constraint of a single, shared geologic history, no discordant datum can be confidently related to another datum, whether it is discordant or concordant."
**We apologize for the wording that led to this misreading of our intention. Our phrasing here is only meant to explain that a single discordant data point cannot be interpreted on its own. That basic concept may not need to be pointed out to the readership of Geochronology.**

As shown from other works seeking to invert discordant data the reality is that discordant data is, more often than not, derived (in your words "related to") from the same geological provenance (e.g. discordant data and concordant data is ultimately derived from terranes that share one or more connected formation ages). This connection can be probabilistically assessed and used in the inversion problem.
**While many zircon datasets may allow for safe use of the assumption that discordant analyses are directly derived from the concordant data, this is still an assumption (for cases where this assumption may not be valid, see Donaghy et al. 2024, Geochronology). In particular, variable discordance may be imposed on different age populations in such a way that any comparison statistical tests aimed at probabilistically assessing goodness of fit are inaccurate (see discussions below regarding the CDC test and the Alta dataset presented here). Responses to later comments show more information regarding this aspect.**

**Even if there is a perfect statistical match between reconstructed discordant and concordant zircon datasets, that is not a firm guarantee that any lower-intercept age derived from such a comparison is valid. We cannot validate geological relationships with statistical tests, we may only invalidate them. All discordant data treatment methods, including the one presented in our work, and as shown by our modeling, are susceptible to various limitations depending upon the**

**assumptions involved in the analysis, and caution is required in the treatment of the statistical approaches here.**

**The revised outline of the introduction would alleviate this particular concern, but this comment helps make the case that our sensitivity and uncertainty modeling test (Section 2) remain in the manuscript to robustly outline the assumptions and limitations of the discordance dating technique, as exposed in the modeling and discussion in those sections of the present manuscript.**

Line 98, the concept of "strict" or not in terms of discordance is a bit nebulous and would be better framed as within or outside the analytical confidence bounds.
**This comment is easily addressed by changing the wording from "strict" to "discrete", which better conveys our original meaning of simply using a discordance filter (e.g., 10%, 20%, etc) when treating U-Pb data.**

Line 101 the paper at this stage in the text now reverts to acknowledge that there are other approaches to invert discordant data. This creates a bit of a non sequitur with what was introduced around line 80.
**Our intention in the earlier sections was to highlight the relative lack of research focus on discordant data, not to give the impression that no other work was done on discordant data. A rewritten introduction as outlined above hopefully satisfies this concern, and we apologize for the way that the original introduction read, it was not our intention. We now fully discuss other discordant data treatment approaches, and have added a full section discussing the outputs of the CDC method in detail.**

Also, the list of works focused on this topic is curiously incomplete. I would have thought that the author would have been familiar with the work on Acasta that uses Pb loss modelling to derive most likely times of fluid-rock interaction?
**We are well aware of the work referred to here (Kirkland et al., 2019; Gondwana Research). It is not cited here because it simply used the CDC method outlined by Morris, et al., 2015, Lithosphere; Olierook, et al., 2021, Gondwana Research, work that is cited. We did not deem it necessary to cite all papers that have used CDC or equivalent methods for investigating discordant data from a variety of locations.**

While the method is contrasted against linear regression approaches (e.g., IsoplotR), other discordance modelling techniques (e.g., isotope diffusion modelling, Bayesian approaches) are not considered. A discussion on how this method compares would strengthen its utility.

Apologies, our meaning here was seemingly not conveyed accurately. We do not 'contrast' our approach with linear regression, we are using linear regression to "benchmark our approach against isochron regression techniques". Meaning, we use linear regression to determine the validity of the discordance dating approach.

We did not include a comparison of the modeling conducted here with the CDC method (which we assume Dr. Kirkland means when mentioning Bayesian methods in this comment) because in our initial testing, the CDC method appeared to be inappropriate for the natural data we present, natural data which we modeled the synthetic data after. This is likely due to several reasons (see revised Section 3.3).

However, Dr. Kirkland is correct that a full discussion of the CDC method strengthens the manuscript and highlights the utility of discordance dating. We now include a comparison of the CDC method as applied to our Alta Tintic formation zircon sample set. The key conclusions are that 1) The CDC method does not resolve the lower intercept age at 24 Ma, and 2) the reason for the CDC method's underperformance is likely to do with the a) reliance on concordant data, and b) the use of K-S or other 'similarity' metrics that rely on not only peak location, but peak height.

This can be seen below. Here we show the outputs from our own calculations that replicate the CDC statistical test. The blue lines are the CDC models applied to the Alta DZ data (71/407 concordant data), with colors corresponding to various discordance cutoffs. Note that we inverted the K-S statistic such that 1 is a perfect match, in order to directionally compare to our discordance dating outputs. No peak is obvious in the CDC data until an unrealistically high discordance filter of 30-40% is used (light blue line), but even then, the peak is small and quite broad.  When using the CDC methodology, and using a Wasserstein distance comparison (Lipp and Vermeesch, 2023), the CDC approach does not resolve any peak in the Alta pluton age range.

A.

[Figure]

B.

Dr. Kirkland's review made us aware of a paper posted on an archive (Mathieson et al.) that was posted publicly after our manuscript was submitted and reviewers had been assigned. That archived manuscript contains code for CDC modeling that allowed us to validate our internal reproduction of the CDC method, which compares well (shown below). Note that the small differences between the CDC method and our own derivation of the CDC method may be due to the Monte Carlo simulations of analytical uncertainty in the Mathieson et al version of the CDC method, small differences in the K-S statistic parameters between our code and that of Mathieson et al., or the fact that our derivation is calculated using Wetherill concordia as opposed to Tera-Wasserburg concordia space.

[Figure]

Importantly, we highlight that we are not showing this to condemn the CDC method. We are simply documenting that the CDC method should not be expected to perform well with the Alta data due to the relative lack (71/407) concordant data points in the dataset.

In fact, the reason that the CDC method does not return accurate or precise chronology results for the Alta stock data requires further investigation. Our analysis suggests that underperformance of the CDC method is partially due to the reliance on comparison tests (K-S test statistics or comparable methods). Results from our testing of the CDC method are shown below.

[Figure]

In this image, the red lines are the 7/6 ages of the concordant data population (all lines are a kernel density estimators using 15 Ma bandwidth). The other curves are CDC projections from various lower intercept points (the color reflects the lower intercept age) through the discordant data points and we show the distribution of those recalculated upper intercept ages. Importantly, the discordance dating age of 24 Ma is shown in the blue curve.

Using a 24 Ma lower intercept, the resulting recalculated data does indeed return correct peak positions, but the relative height of those peaks does not closely match the concordant data point age distribution, hence the relatively poor comparison metrics. The K-S statistic accounts for differences in relative peak heights as well as positions, so the red and blue curves do not yield highly comparable distributions when using that comparison statistical test.

An improved CDC method that simply compared peak locations between concordant and modeled upper intercepts could possibly remove any potential biases induced by preferential discordance imposed on different populations. For instance, perhaps the 1.1 Ga zircon peak in the Tintic formation was more immune to discordance at 24 Ma than the 1.4 and 1.8 Ga zircon populations. This could be simply due to less radiation damage or some compositional difference in source terrains (higher U or Th in the source rocks for older grains). Thus, the relative peak heights of discordant data would be higher for those older populations than the 1.1 Ga population. Using a peak matching algorithm would decouple the reconstructed peak locations from possible biasing of the peak

**heights and perhaps make the CDC test more reliable for samples like the Tintic formation detrital zircons discussed above.**

**Based on this comment by Dr. Kirkland, we have added a new Section 3.3 (replacing the text recommended for removal by Drs. Ickert and Kirkland), which outlines the results discussed above and includes a single figure discussing the CDC approach.**

The description of the Concordance-Discordance-Comparison technique (and its inverse in Olierook) is incomplete, as it is not merely a projection method. Instead, it employs a Bayesian Monte Carlo approach to probabilistically evaluate all possible Pb loss events, ultimately determining the most likely Pb loss age (or the primary crystallization age in the inverse application).
**We apologize for the apparent incompleteness. It would be useful if the appropriate reference was mentioned here, as we cannot determine how either Bayesian (i.e., one the generates and interrogates a posterior distribution which is generated by combining the likelihood distribution and the prior distribution), or Monte Carlo approaches are employed in the CDC method as currently published.  We do see that Monte Carlo approaches are used to model uncertainty in an archived manuscript in prep (Mathieson, Kirkland, et al) which was not public at the time we submitted our manuscript.**

Line 108, the work states a "safe" assumption is that "all the zircon grains have a shared thermal and geological history" yet is this really the case. Specifically, other works have clearly demonstrated (that in some cases) the susceptibility of the zircon cargo to post crystallization modification is highly variable. This is easily demonstrated by considering the heterogenous alpha dose (or U content) within any detrital zircon population. Effectively, this means that in many situations, components within a chemically heterogeneous zircon population will be "blind" to certain events whereas other grains may record that event.
**This is exactly the scenario that, as shown above, may lead to the underperformance of the CDC method in the Alta DZ data presented here. There is a mismatch between the relative heights of concordant data, and reconstructed discordant data upper intercepts.**

This assumption, that all zircons will respond to the same event, at least needs to be discussed and considered as it has fundamental implications for when the proposed methodology would be the most effective or not.
**This is similar to a comment on the wording raised by Dr. Ickert. It would be straightforward to add some discussion of the kind requested here and in**

**previous comments by Dr. Ickert. Specifically, by using "thermal and geological" we are referring to events that affected the entire rock sample in question – metamorphism, heating, pluton emplacement, etc.). We do not mean specific processes that could variably affect individual zircon growth zones differently. We will reword this to make this aspect of our meaning clear.**

Line 120-122 is a statement I would certainly agree with and has been said various times before, so you probably could support your point with references.
**Good point. Our revised introduction will include relevant references to other work that has taken different approaches to address the same issue.**

"…..enables geochronologists to extract meaningful geological information from discordant datasets, turning previously discarded data into valuable insights" Mathieson et al., 2024. Turning Trash into Treasure: Extracting Meaning from Discordant Data via a Dedicated Application. G4 in press, 2024. DOI: 10.22541/essoar.173315682.28715367/v1.

"CDC modelling of discordant U-Pb zircon analyses may provide a means to recognise the distal footprint of otherwise difficult to date tectonothermal events and extract useful information from often discarded analyses."
https://doi.org/10.1016/j.gr.2017.08.004
**The first quote comes from a manuscript that was posted on an open archive (Nov. 27th, 2024) after the date of submission and assignment of the reviewers of our manuscript (Oct. 28th, 2024).**

The work may be improved by considering that discordance in the case study dataset could also relate to physical mixtures (e.g. core-rims).
**This was discussed in the manuscript in Section 3.3, but this section is likely to be removed from the text based on the comments made by Drs. Ickert and Kirkland.**

The text on many of the figures is illegible (this may just be an issue with the review pdf but the scaling of text especially in figure 2 needs to be made more consistent across the different components of the figure).
**This is indeed an issue with the PDF and will be addressed at the manuscript typesetting stage.**

Line 294, it would be more informative to know how it performs relative to Sharman, G. R., & Malkowski, M. A. (2024) and the Concordance-Discordance-Comparison test, rather than against linear regression approaches which clearly are not designed to deal with such over dispersion.

**We will add discussion of the Sharman and Malkowski paper in the revised and shortened introductory text, and a full comparison to the CDC test is outlined above and will be included in a revised version of the manuscript.**

It would be useful to have a more complete consideration / discussion of how uncertainty in the inversion method has been dealt with. Presumably this is a function of the step size the trial cords have been spaced at?

**This extra discussion would be straightforward to add. The summed likelihood method employed here is naïve to the underlying data structure (it is not an inversion), and the only impact the node spacing has is to refine the shape of the underlying likelihood curve. Wider node spacing yields a less-smooth likelihood curve shape. Uncertainty is calculated by our bootstrapped resampling approach (Fig. 7), and after Dr. Ickert's suggestion, now also estimated including decay constant uncertainties.**

The method relies on a summed-likelihood approach, but it is unclear how it handles data gaps, outliers, or clustering effects. Could certain grain populations disproportionately influence the results?

**We address the uncertainty by using a bootstrapped resampling method to address the sensitivity of the discordance dating to data distributions. The model simply reports distributions of data across Pb/U space, which is then left to the geochronologist to interrogate and interpret. Our sensitivity analysis is important in this regard, as it exposes some of the limits of the discordance dating approach, such as the possibility of generating artificially old lower intercept ages. We attempted to make that point clear in Section 2.2, but have added text to more clearly convey this point.**

Line 265-266 "and therefore yields more accurate results for complex datasets" more accurate than what? Linear regression? Well obviously, it must, as it is designed to account for dispersion in the data.

**As indicated by the text in the introduction to this section, we are merely using linear regression functions to "benchmark our approach against isochron regression techniques". We are using the well accepted linear regression methods to perform an assessment of the sensitivity and accuracy of the discordance dating methods. We did not mean to convey that they are competing methods and have reworded this text to more clearly convey this point.**

More accurate than the other Pb loss modelling approaches?

**As shown in response to a previous comment by Dr. Kirkland, yes. We now highlight this fact in a revised version of the manuscript by including a discussion of alternative techniques applied to the Tintic formation detrital zircon data.**

I would be rather confident in guessing that the answer to that question depends entirely on the underlying geological controls on the dispersion e.g. chemically heterogeneous grains (with heterogeneous age) variably responding to different episodes of Pb loss or chemically homogeneous grains (with heterogeneous ages) undergoing a single phase of Pb loss. The assumption that all zircons in a sedimentary unit share a common post-depositional history is not universally valid.

**This is addressed in a reply to a comment by Dr. Ickert. Our discordance dating approach relies on variable response to a discordance-inducing event. This may not be the case for other methods focused on analysis of discordance detrital zircons.**

Localized alteration, differential Pb mobility, or variable zircon radiation damage could result in multiple resetting events rather than a single event. How does the method account for this?

**We did not intentionally imply that each grain has experienced an identical chemical history. In fact, our model relies on a variable response (among the entire grain/growth zone population) to, in this case, an apparently single discordance-inducing event. This creates the spread in discordance and results in the ability to apply things like discordance dating to date discordance-inducing events. With the Alta dataset, there is a clear single discordance-inducing event, so our model only outputs a single event, because it is simply mapping probability distributions in Pb/U isotopic space. More complex scenarios are considered in Reimink et al., 2016, and the interested reader is directed there for more details on this aspect. As pointed out by Dr. Ickert, the discordance dating presented here is a modified version of the method outlined in that work, and following his suggestion we do not include further discussion of that model apart from the introductory text already in the manuscript.**

The model assumes that the youngest discordant grains define the resetting event. However, zircon discordance can result from multiple overlapping processes (fluid mobility, radiation damage, Pb clustering). It would be nice if the paper could clarify how, it differentiates true geologic resetting from more complex Pb-loss mechanisms.

**Our discussion in Section 3.3 in part covered this problem, focusing on the structural mechanisms of discordance. However, this section has been cut to shorten the manuscript to a technical note (see comments by Dr. Ickert on this as well).**

**The model, like most statistical methods, models U-Pb data alone and cannot rigorously evaluate the causes of discordance. Radiation damage does not itself induce discordance. Chemical disturbance of radiation damaged lattice domains can, however, cause discordance. Pb clustering has only rarely been documented and often produces spuriously old ages in very ancient grains. Other mechanisms**

**for producing discordance, as outlined and summarized in the present manuscript, have a series of shared geologic 'forcing' processes (metamorphism, fluid ingress, thermal perturbation, etc.).**

**Our goal is to produce a statistical method that allows geochronologists to more fully interpret U-Pb datasets, but ultimately it is simply a statistical model that necessarily leaves the geological interpretation up to the geochronologist. We feel that this is the appropriate approach, as additional information (grain structural information, trace-element data, alteration histories of a particular rock sample, other thermochronology data) can be leveraged into addressing the questions posed in this comment.**

While synthetic datasets were tested, real-world zircon populations may exhibit more complex discordance patterns than the simplified scenarios presented. A more robust sensitivity analysis including mixed multi-stage alteration histories could improve confidence in the method.

**The synthetic datasets presented here really serve only one purpose – to test the limits of our method when applied to a single discordance-event population, closely mirroring the geological scenario captured by the Alta DZ dataset. Our intention was to test the sensitivity and accuracy of the age information derived from this discordance dating to robustly evaluate the uncertainties. Therefore, we prefer to keep the synthetic data closely resembling the Alta DZ dataset, in part to limit the length of the manuscript. The most important outcome of the modeling is to show that artificially old ages can be created, and that discordance dating is very sensitive to the position of the most discordant analysis. These results help guide the reliable interpretation of results as well as direct future analytical approaches to using discordant data.**

A test case where the method is applied to a sample with known independent constraints (e.g., metamorphic zircon rim ages) would validate its accuracy.

**It is possible that we don't completely understand this comment, but it appears that the requested test is exactly the test we have conducted using the Alta Stock alteration halo. These Tintic detrital grains have experienced a known resetting event with well-understood age distributions, which are accurately reproduced by the discordance dating method. The age of the Alta stock and surrounding metamorphic aureole are well known, as shown in the papers reference in the present work.**

The discussion on whether Pb loss is due to fluid infiltration versus recrystallization is a bit speculative without clear microstructural evidence (e.g., TEM, Raman spectroscopy).

Suggest incorporating or citing complementary methods that could distinguish these processes.
**We agree with this comment regarding discordance (not only Pb-loss but possible U-gain or other mechanisms for creating discordance), and were attempting to make exactly this point in the text. However, this section will likely be cut in the next version of the manuscript following comments by Dr. Ickert.**

The analytical dataset appears to be of generally high quality, although I note the 207/206Pb values for glass NIST 612 appear a bit low.
**This was noted by Dr. Schmitt and is addressed in the response to that comment.**

---

## Author Response (AR2)

**We thank Dr. Kirkland for the additional comments and concerns that help clarify the manuscript's writing.**

I commend the authors for their thoughtful and thorough revisions in response to the initial review. The manuscript now presents a clearer and more impactful contribution to the methodological advancement of discordance modelling in U–Pb geochronology. Well done!

The manuscript generally offers a reasoned and balanced evaluation of the methodology, especially in regard to the improvements introduced by the authors. However, I recommend a few minor textual refinements to enhance clarity and precision, particularly in articulating the limitations and applicability of the proposed approach.

**While we appreciate the concerns raised below by Dr. Kirkland, many of the comments in this round of revisions focus on the CDC method instead of the discordance dating method that is presented, benchmarked, and evaluated with natural data in the present manuscript. We do not completely agree with some of the framing of the CDC method outlined in these comments, but a full evaluation of all discordance approaches is well outside of the scope of the present manuscript. We wish to avoid, as much as possible, detailed analysis and discussion of that method, its variations, and the Reimink et al. 2016 construction (and its variations). Thus, we have generally addressed the below concerns by removing text that went further into the discussion of the CDC method than we would prefer, thereby alleviating the majority of the concerns raised in the minor comments provided below. All other concerns have been changed according to Dr. Kirkland's wishes.**

Line 816: The current phrasing may unintentionally overstate the general superiority of the proposed method relative to the CDC approach. As the authors have previously noted, the CDC test is well-suited to datasets exhibiting complex Pb loss, particularly those containing concordant components. In contrast, the method presented here appears to perform best when applied to datasets with well-defined, single-component Pb loss trends (e.g. to a single time), whether or not they include concordant analyses. I recommend rewording this section to more clearly qualify that the improvement pertains to a specific class of data. Additionally, the authors might wish to acknowledge that with further development, the Reimink et al. method could potentially be adapted to better handle cryptic Pb loss. A more nuanced comparison would better reflect the diversity of datasets encountered in geochronology.

**We have changed the wording in Line 816 to remove any unintended bias in our evaluation of the different methods.**

KS test vs. peak fitting: The discussion on the performance of the KS test versus peak-fitting methods would benefit from acknowledging that the optimal approach is highly dependent on the data structure. Specifically, peak fitting may be preferable when the dataset exhibits clearly resolved trends, while the KS test may offer advantages in identifying multiple events or more cryptic distributions. A brief clarification of this point would help guide users in choosing the most appropriate analytical path.

**To alleviate this concern we have chosen to remove the sentence that explicitly suggested using peak finding methods within the CDC method. As highlighted above, a full evaluation of other discordance approaches is outside the scope of the present manuscript. We have kept the limited discussion of the CDC method in Section 3.3, however, we wish to keep the discussion of the different methods at relatively superficial level because the goal of this manuscript is not to evaluate all options and their variations, it is to present and benchmark a new approach.**

Figure 10A: I note that in my own analysis using a different underlying data distribution, I reproduced the inverse result to that shown in Figure 10A, with the CDC method more effectively recovering the expected geological signal. This observation reinforces the point that the dominant source of scatter in a given U–Pb dataset, be it due to discrete Pb loss events or complex, cryptic discordance will dictate which method yields the most geologically meaningful result.

**We are not sure how to address this comment without access to the mentioned different data distribution. However, we prefer to leave a full evaluation of the reliability of other discordance modeling methods to future work.**

Line 100: References are needed to support the claim made here.

**These references are included in the paragraphs following this line, so we have simply added "as discussed below" to point the reader to the references outlined in the revised introduction.**

Line 897: When referencing the "Tintic zircon data," it would be helpful to briefly and in words characterize the dataset's key features (e.g., some few words spent on the pattern of discordance and degree of Pb loss). Doing so would aid readers in assessing the generalizability of the method to other datasets with different characteristics.

**We have added a clause stating "..., which experienced a discrete overprinting event ca. 30-24 Ma" to this sentence to clarify the data distribution.**

Literature context (e.g., Kirkland et al. 2020): While I appreciate that citation decisions are at the authors' discretion, I would again gently suggest that omitting comparisons to

published studies where the CDC test has proven more effective (e.g., cryptic Pb loss in the Acasta gneiss) may give the appearance of a selective literature treatment, which I am sure is unintentional. Including such examples rather than weaking the findings in this study would rather further reinforce the conclusion that method performance is inherently data-dependent and would provide a more balanced context for readers.

**We do not conclude that the performance of various methods is data dependent and neither do we agree with some of the conclusions outlined in Kirkland et al. 2020 regarding the performance of both classes of discordance modeling approaches. In the present work we do acknowledge that there is much work to be done to fully and rigorously evaluate the various classes of discordance treatment approaches, and their variations, across a range of geological settings (Line 914). However, to avoid any perception of selective literature treatment, we have included a citation of Kirkland et al. 2020 in the section introducing the CDC method.**

Line 883; but some of the KS tests at higher % discordance cut off's do appear to return the 24 Ma age. Some minor editing is needed for accuracy.

**We have added the clause ", except for minor, <24 Ma peaks when using very high (<30%) discordance filters" to this sentence to maintain accuracy.**

Line 886; lack of resolution in a specific data distribution case.

**We have changed this sentence to read: "This suggests that part of the reason for the lack of lower intercept age resolution for the Tintic formation detrital zircon data returned…" to make it more explicit that we are referring to the data provided in this manuscript.**

Line 908; Likewise multiple Pb loss episodes would impart dramatic biases in reconstruction methods based on linear arrays.

**We are not clear as to what is referred to by "linear arrays". Both the CDC method and the discordance dating approach use linear recalculations (calculations along linear chords in U-Pb space). These two approaches are very similar in that regard, with the main difference being in the evaluation of such reconstructed data (reconstructed ages compared to concordant data in the case of the CDC, and mapping of probability distributions independent of concordant data in the case of the discordance dating). Regardless, a full evaluation of the various merits of all classes and variations of discordance approaches is outside of the scope here.**

In summary, I consider the manuscript nearly ready for publication. With these minor textual clarifications and an emphasis on maintaining consistency in the framing of

method applicability, it will serve as a valuable resource for the geochronology community. Thank you for giving me the opportunity to provide feedback which I hope has helped the authors.

**These comments have helped clarify the manuscript and we thank Dr. Kirkland for the additional revision.**

Sincerely, Chris Kirkland, Perth WA